# Platanus-allee is a de novo haplotype assembler enabling a comprehensive access to divergent heterozygous regions

Rei Kajitani[1], Dai Yoshimura[1], Miki Okuno[1], Yohei Minakuchi[2], Hiroshi Kagoshima [3], Asao Fujiyama[3], Kaoru Kubokawa[4], Yuji Kohara[3], Atsushi Toyoda [2,3] & Takehiko Itoh [1]

The ultimate goal for diploid genome determination is to completely decode homologous chromosomes independently, and several phasing programs from consensus sequences have been developed. These methods work well for lowly heterozygous genomes, but the manifold species have high heterozygosity. Additionally, there are highly divergent regions (HDRs), where the haplotype sequences differ considerably. Because HDRs are likely to direct various interesting biological phenomena, many genomic analysis targets fall within these regions. However, they cannot be accessed by existing phasing methods, and we have to adopt costly traditional methods. Here, we develop a de novo haplotype assembler, Platanus-allee (http://platanus.bio.titech.ac.jp/platanus2), which initially constructs each haplotype sequence and then untangles the assembly graphs utilizing sequence links and synteny information. A comprehensive benchmark analysis reveals that Platanus-allee exhibits high recall and precision, particularly for HDRs. Using this approach, previously unknown HDRs are detected in the human genome, which may uncover novel aspects of genome variability.

[1] School of Life Science and Technology, Tokyo Institute of Technology, Meguro-ku, Tokyo 152-8550, Japan. [2] Comparative Genomics Laboratory, National Institute of Genetics, Mishima, Shizuoka 411-8540, Japan. [3] Advanced Genomics Center, National Institute of Genetics, Mishima, Shizuoka 411-8540, Japan. [4] Ocean Alliance, The University of Tokyo, Bunkyo-ku, Tokyo 113-0033, Japan. Correspondence and requests for materials should be addressed to T.I. (email: takehiko@bio.titech.ac.jp)

In diploid genomes, the distribution of heterozygosity generally deviates from the Poisson distribution[1], and highly divergent regions (HDRs), in which haplotype sequences differ considerably, occasionally > 5%, have been observed[2–12]. HDRs were reported to be associated with certain biological phenomena, such as marked variations of morphs[2,6,7], social behavior[8], speciation[9,10], sex-determining systems[11], and gametic incompatibilities in plants[12]. These HDRs have been investigated employing laborious and costly procedures such as fosmid/BAC cloning and sequencing[2,4,12] of targeted regions, Sanger/ 454 sequencing methods (with relatively long reads but low throughput)[3–5], analysis of many individual genomes[6,7,9–11] and sequencing of a trio (father, mother, and offspring)[13].

Whole-genome and high-throughput sequencing data were utilized as a cost-effective method in an attempt to detect variants. However, a majority of the tests of variant-detections and phasing (detection of linkages between variants) was performed primarily for the human genome with relatively rare HDRs (heterozygosity ~0.1%)[14–17].

FALCON-Unzip[18], the phasing tool for the single-molecule DNA sequencing, succeeded in constructing haplotigs (heterozygous phased blocks) with large contig-N50 (6.92 Mbp) for *Arabidopsis thaliana* F1-hybrid (estimated heterozygosity, ~1%). However, the value for the more heterozygous plant, grape, was reported to decline to 0.779 Mbp. According to the supplementary analysis and surveys in ref. [18]., there are certain effects to phasing performances from heterozygosity, and half of the other species surveyed (6/12) were estimated to have heterozygosities > 1%, implying that there is room to develop versatile haplotype phasing methods.

These observations mainly arise from the two major technical issues. First, de novo assembly itself is difficult for highly heterozygous organisms[1,3–5] when using conventional procedures. Second, read mapping-based variant detections were often hampered owing to the size of the haplotype differences, which hinders their mapping onto the assembled genome[1]. Due to these difficulties, it is possible that many elements in HDRs and highly heterozygous genomes were not observed in previous studies.

Consequently, the development of a phasing tool able to handle these HDRs is an important requirement for the comprehensive survey of diploid genome diversity, and it can help reform current single nucleotide polymorphism (SNP)-centered genetic frameworks. To this end, we develop a de novo haplotype assembler, Platanus-allee, which constructs each haplotype in a diploid genome independently without conventional procedures of consensus-sequence assembly and variant-calling. Benchmarks targeting the organisms with the wide-range of heterozygosities (0.1–3.5%) illustrates the versatility of this tool and its advantage to assemble haplotypes of HDRs.

## Results

**Development of Platanus-allee**. The basic algorithm of Platanus-allee is based on the arrangement of two independently assembled sequences derived from each haplotype of the corresponding two homologous chromosomes (Fig. 1a), in contrast to the most haplotype assemblers, which try to split the consensus sequences into two homologous regions (for details, Supplementary Note 7). These concepts were mainly realized by adding the information of de Bruijn graphs[19] to the following two modules: scaffolding and gap-closing modules of the Platanus genome assembler[1]. The feature of Platanus-allee is that it never constructs consensus sequences by multiple alignment or removal of one side of bubble structures in graphs before finishing phasing.

Platanus-allee exactly distinguishes the $k$-mers in reads (typically, $k > 100$ at the final step of contig-assembly), possibly reducing the number of errors that occurred in the omitted steps. To apply this function, accurate reads with low error rate (<1%) are essential. Currently, the suitable data type is the Illumina short-reads and Platanus-allee is designed to primarily use these data. Long-reads with high error rate (>10%) can be optionally utilized in the downstream steps. The assembly step of Platanus-allee is supported by the following two strategies: (1) untangling complex graph structures and (2) skipping difficult regions by scaffolding. The former can generate gap-less sequences but may be hindered from extending by complex and repetitive regions, in contrary to the latter. The two complementary functions are iteratively applied to long and gap-less phased blocks (Fig. 1b).

Platanus-allee requires at least one library from the Illumina sequencer and can exploit mate-pair (jumping) libraries, linked-reads[17] (barcoded reads from long DNA fragments, 10X Genomics), and single-molecule long reads (Pacific Biosciences (PacBio) or Oxford Nanopore, utilizing Minimap2[20]). Note that, although Illumina reads are short compared to single-molecule reads, their accuracy (>99% for each base) may help to effectively distinguish homologous haplotypes in some cases. Essentially, Platanus-allee is a hybrid assembler that primarily uses the accurate short-reads and can combine other types of libraries to enhance performances.

Initially, Platanus-allee attempts to untangle cross structures, such as X-like forms, using link information from single reads, paired-ends (PEs), mate-pairs (MPs), and linked-reads mapped to both de Bruijn and scaffold graphs. The cross structure consists of one central node and four remaining nodes possibly representing a homozygous sequence and heterozygous sequences, respectively. When there is adequate link information between the heterozygous nodes, the duplication of the central node and the assembly of two sequences (untangling) correspond to the phasing of heterozygous variants (see "Methods" section). The untangling processes are iteratively executed for both de Bruijn and scaffold graphs (Fig. 1b). Similar ideas for untangling have been previously proposed[19,21,22], and our improvements consist of an application to the two types of graphs and handling of various types of libraries.

Furthermore, to obtain the continuous phased homologous sequences correctly and to improve structural accuracy, Platanus-allee employs an algorithm called haplotype synteny-based assembly (Fig. 1a; see "Methods" section). This was designed using the following assumptions for a diploid genome: (1) both HDRs and low-heterozygous regions exist and are distributed in patchwork structures, and (2) chromosome-scale synteny is conserved between a homologous chromosome pair even if heterozygosity is high and the local structural variants are contained. After the anchor bubble detection and alignment of homologous haplotype sequences (see "Methods" section), edge regions that are not aligned to the counterparts (*i.e.*, non-syntenic) are determined. The boundary between the aligned and unaligned regions is assumed as the mis-assembly point, and the haplotype sequences are divided according to the boundary. Divided sequences are re-extended through the following iterations to determine the correct structures.

The procedure consisting of untangling, scaffolding, correction, and gap-closing is iterated twice in the default setting, and finally, the output phased blocks are obtained (Fig. 1b). The format of the block is similar to "megabubble style" of Supernova[23], the de novo assembly-based phasing tool for linked-reads, in which a heterozygous haplotype sequence is associated with a homologous counterpart from end to end as a bubble. For each bubble, the sequence with more non-N bases is defined as a primary-bubble, whereas the other is defined as a secondary-bubble. Optionally, primary-bubbles and non-bubble sequences can be connected, obtaining the conventional consensus format of scaffolds, which

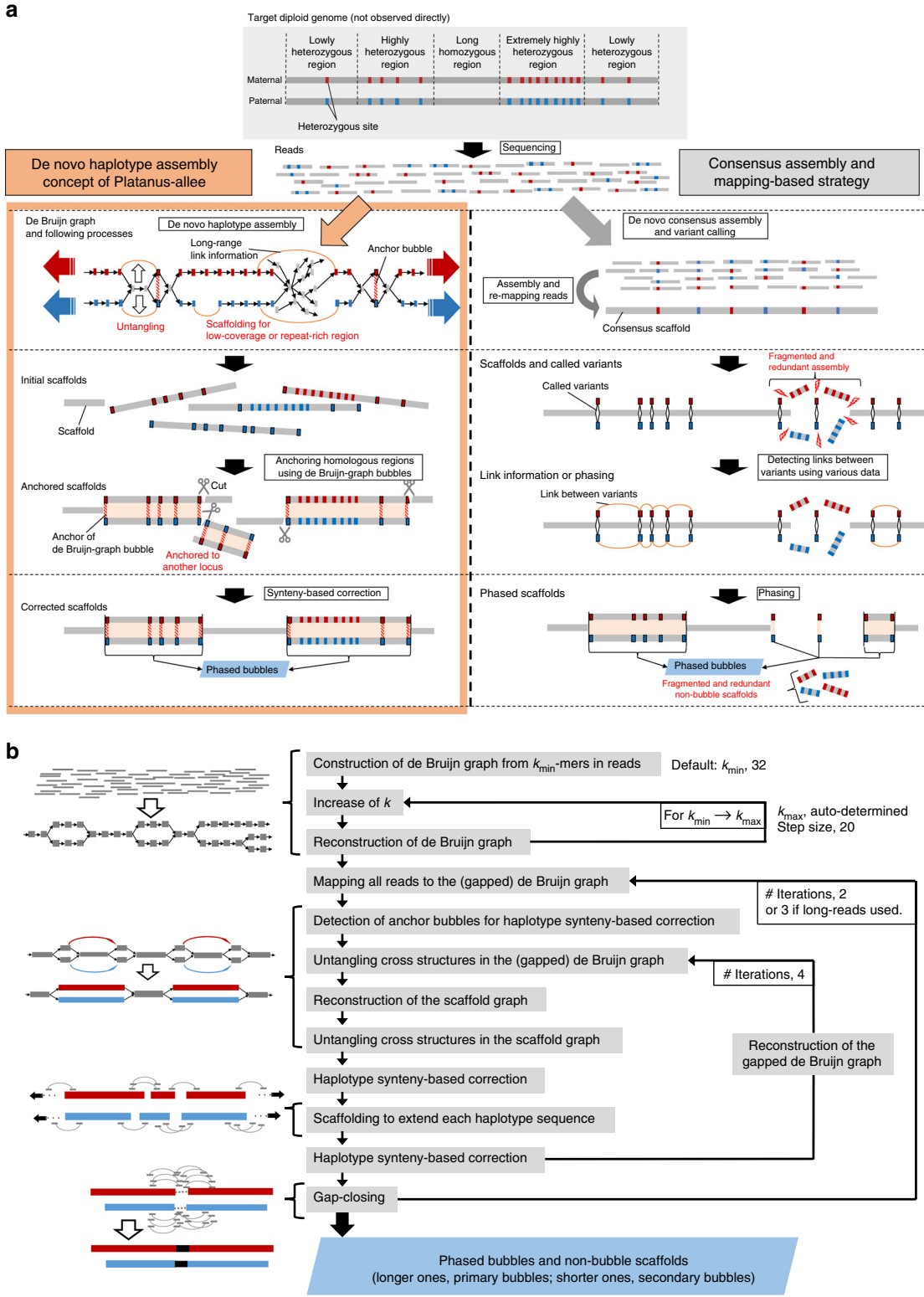

**Fig. 1** Platanus-allee algorithm. (**a**) Schematic model of the concept of phasing and haplotype synteny-based assembly. (**b**) Workflow

may be more contiguous for highly heterozygous organisms compared to the results of other assemblers[1].

**Benchmarks for data of highly heterozygous organisms**. We benchmarked the contiguity, coverage, and accuracy of the phased block generated by de novo assembly-based phasing tools,

namely Platanus-allee (version 2.0.2), FALCON-Unzip (version binary from 11/02/2017, applied in ref. [13]), and Supernova (version 2.0.0). In the original articles[18,23], the latter two recorded the large N50s. Therefore, we used them as cutting-edge tools. Below NG50, length such that 50% of the genome size contained in sequences of this value or greater is used as the indicator of contiguity of assemblies. LG50 is the number of sequences whose

lengths $\geq$ NG50. NG50 can be measured for both scaffolds (sequences containing gaps (Ns)) and contigs (gap-free sequences). In this study, scaffold-NG50 equals contig-NG50 if an assembled sequence has no gaps, such as the FALCON-Unzip's result. The targets of benchmarks were the phased blocks, so that the primary contigs of FALCON-Unzip, the mosaics of haplotypes, were divided at boundaries of blocks (see "Methods" section). The results of Platanus-allee and Supernova were used as bubble and non-bubble sequences, or, in the "megabubble style".

To optimize the existing tools, multiple parameter sets were tested, and we selected a result with the largest scaffold-NG50 of phased blocks for each tool. For FALCON-Uuzip, the four parameter sets consisting of the ones used in the previous studies[13,15,26] or the distributed one (Supplementary Note 12; Supplementary Table 6) were tried. Illumina-reads-based polishing tool, Pilon[24], and redundancy-reduction tools, Purge-Haplotigs (PH)[25] were both applied and the results were also benchmarked. For Supernova, either all reads or reads downsampled to reach $56\times$ of expected coverage depth (optimum value in the manual) were tested.

Finally, in addition to the phased blocks, consensus sequence statistics were evaluated. Note that Platanus (version 1, the base of Platanus-allee) can only construct consensus scaffolds, not phased blocks, and its results were exclusively included in the table for consensus sequences.

We initially generated the sequencing dataset of the swallowtail butterfly, *Papilio polytes*, using various platforms: Illumina PEs, MPs, PacBio long reads, 10X Genomics linked-reads (10X) (Supplementary Table 1). According to the $k$-mer frequency distribution of PEs, high heterozygosity was represented (Supplementary Fig. 1) and the estimated genome size was 240 Mbp. To quantify heterozygosity, we also applied the $k$-mer analysis tool, GenomeScope[26], which can estimate that value, to all species used for benchmarks in this study (Supplementary Table 5), resulting in 1.52% heterozygosity for this sample. This species has two distinct wing patterns and these phenotypes are determined by a single locus[2], which is an HDR, including not only a large inversion ($\sim$130 kbp) but also a low sequence identity (<95%) between two haplotypes[2].

The contiguity statistics of phased blocks for *P. polytes* were measured (Table 1; Supplementary Table 7). The consensus sequence statistics and the run times of Platanus-allee under the condition of 20 threads and are shown in Supplementary Table 8–10. We expected that Platanus-allee would be efficient in low-coverage long-reads datasets, and therefore the input PacBio data were reduced to $20\times$. Contiguities of sequences were measured using NG50 and LG50. For the various combinations of the input data, Platanus-allee exhibited the largest scaffold-NG50 of phased blocks compared to other tools. Remarkably, FALCON-Unzip and Supernova no longer achieved mega-scale NG50, indicating their weaknesses in the analysis of a highly heterozygous organism. To measure the quality of the phased blocks, we mapped Illumina MPs (insert-size, 15 kbp) to the blocks, and counted the number of pairs with reads exactly matching the same block. For the accurate and contiguous phased blocks, we expected to obtain a large number. The evaluation tool that uses universal single-copy orthologs, BUSCO[27], was also utilized, and the results (Table 1; Supplementary Table 7) confirmed that the quality of Platanus-allee phased blocks was higher than those of the other tools.

We examined the HDR responsible for the wing pattern, $H$ locus, on the phased blocks. Platanus-allee was used for the construction of the bubble that covers the $H$ locus entirely (Fig. 2a). The structure of the inverted HDR was consistent with that reported in a previous study[2], which was confirmed using different approaches. Using FALCON-Unzip, we also obtained the phased haplotype sequences of $H$ locus as a bubble, but the flanking regions were fragmented compared to the Platanus-allee (Fig. 2b). The Supernova output was also represented as non-bubble sequences for $H$ locus. These were non-redundant and one of the haplotypes may be lost or finely fragmented, and therefore our analysis demonstrated an example of a biologically important HDR plus the mega-scale flanking regions that can be examined using Platanus-allee.

For the second benchmark, we generated the data from the amphioxus *Branchiostoma japonicum* using a similar library configuration as for the first benchmark (Supplementary Table 2). Similar to the other reports on amphioxus[3], this sample had considerably higher heterozygosity than *P. polytes*, inferred from the $k$-mer frequency distribution (Supplementary Table 5; Supplementary Fig. 1b), with an estimated genome size of 390 Mbp. As a result, the contiguities (scaffold-NG50 and scaffold-LG50) of Platanus-allee phased blocks outperformed those of other tools (Table 1; Supplementary Table 7). Compared to the results obtained for the analysis of *P. polytes*, the scaffold-NG50s of phased blocks of Platanus-allee with MPs were much larger than those without MPs, which may indicate that the effectiveness of MP increases with heterozygosity.

To evaluate the recall and precision of phased blocks in detail, we additionally generated the Moleculo synthetic long-read library[28], consisting of the contigs locally assembled for each DNA fragment ($\sim$10 kbp). Although expensive to construct, this library was expected to have accurate sequences, and therefore, it was considered suitable for the evaluation. Here, the Moleculo contigs or the phased blocks were divided into non-overlapping and fixed-length (1 kbp or 5 kbp) fragments and counted the exact matches to the other set. The indicators were defined as follows: recall, number of Moleculo-fragments matched to phased blocks/number of all Moleculo fragments; precision, number of phased block-fragments matched to Moleculo contigs/number of all phased block fragments. As the integrated one, F-measure (harmonic mean of recall and precision) was introduced.

These metrics depend on the exact matches and a fine-scale accuracy is required to achieve excellent results. Consistently with the contiguity, Platanus-allee with MP outperformed FALCON-Unzip and Supernova for both recall and precision (Fig. 2c; Supplementary Table 11). Against the combination of FALCON-Unzip, Pilon, and PH, Platanus-allee was inferior for recall but superior for F-measure. FALCON-Unzip produced a large total size (917 Mbp; with Pilon and PH, 978 Mbp), but it exceeded the $2\times$ estimated genome size (780 Mbp) and the precisions were low, indicating many false positives. Considering that the amount of input PacBio reads was large ($156\times$) for FALCON-Unzip, the test of full input (coverage depth $> 100\times$ for both PE and MPs; Supplementary Table 2) to Platanus-allee was performed, and it could exceed the other tools for both recall and precision (Supplementary Table 11). Overall, it covered true haplotype sequences comprehensively with less false positives.

For users of phasing tools, the ideal outputs of all heterozygous genomic regions are recognizable data structures such as "bubbles". We prepared heterozygous sequence pairs from the Moleculo datasets and examined the relationships between heterozygosity and the fraction of pairs that were correctly phased as bubbles for each tool. Specifically, each Moleculo contig was aligned to the others and an alignment corresponding to an alternative allele was detected based on the best hit among non-exact hits, which most likely contained heterozygous variants. The resultant alignments were divided into approximately 1 kbp-blocks and the heterozygous "1k-mer pairs" for evaluation were constructed (see "Methods" section). For each phased block set, we counted 1k-mer pairs that had exact matches for both sequences (phased). Furthermore, for each 1k-mer, we determined

**Table 1 Phased block statistics**

| Species | Assembler | Input data | Total (Mbp) | Bubble total (Mbp) | Bubble-total / genome-size | Scaffold NG50 (kbp) | Scaffold LG50 (#) | Contig NG50 (kbp) | Contig LG50 (#) | % gaps | BUSCO duplicate complete (%) | % exact-match MP15k pairs |
|---|---|---|---|---|---|---|---|---|---|---|---|---|
| P. polytes | Platanus-allee | PE + 4 MP | 442 | 393 | 0.818 | 404 | 344 | 102 | 1,282 | 2.80 | 79.03 | **35.83** |
| | | PE + 4 MP + PacBio (20 ×) | 473 | 456 | 0.950 | **3,225** | **51** | 161 | 868 | 2.79 | **89.15** | 33.82 |
| | | PE + 4 MP + 10X | 449 | 407 | 0.848 | 698 | 207 | 113 | 1,214 | 2.60 | 83.25 | 35.61 |
| | | PE + 4 MP + PacBio (20 ×) + 10X | 476 | 460 | **0.959** | 2,392 | 65 | 143 | 987 | 2.71 | 88.94 | 33.57 |
| | FALCON-Unzip | PacBio (99 ×) | 481 | 404 | 0.843 | 413 | 352 | 413 | **352** | **0.00** | 70.68 | 31.10 |
| | FALCON-Unzip, Pilon, PH | PacBio (99×) + PE | 471 | 422 | 0.880 | 421 | 353 | **421** | 353 | **0.00** | 77.48 | 34.60 |
| | Supernova | 10X | 313 | 122 | 0.253 | 79 | 789 | 31 | 2,362 | 1.92 | 24.32 | 29.98 |
| B. japonicum | Platanus-allee | PE + 3 MP | 720 | 686 | 0.880 | 1,090 | 194 | 47 | 4,406 | 3.79 | 86.30 | |
| | | PE + 3 MP + PacBio (20 ×) | 739 | 715 | 0.916 | 1,514 | 142 | 48 | 4,300 | 4.40 | **87.53** | |
| | | PE + 3 MP + 10X | 732 | 694 | 0.889 | 1,155 | 172 | 33 | 6,271 | 3.94 | 84.15 | |
| | | PE + 3 MP + PacBio (20 ×) + 10X | 750 | 720 | **0.923** | **1,516** | **140** | 34 | 6,124 | 4.53 | 85.28 | |
| | FALCON-Unzip | PacBio (156×) | 918 | 378 | 0.484 | 172 | 1,179 | 172 | 1,179 | **0.00** | 74.34 | |
| | FALCON-Unzip, Pilon, PH | PE + PacBio (156 ×) | 978 | 852 | 1.092 | 1,075 | 162 | **1,075** | **162** | **0.00** | 80.98 | |
| | Supernova | 10X | 697 | 177 | 0.227 | 18 | 8,779 | 11 | 17,306 | 2.98 | 41.41 | |
| C. elegans | Platanus-allee | PE + 3 MP | 195 | 179 | 0.897 | 470 | 112 | 43 | 1,242 | 4.89 | 77.09 | |
| | | PE + 3 MP + PacBio (20 ×) | 205 | 198 | **0.988** | **902** | **64** | 60 | 992 | 4.95 | **86.05** | |
| | FALCON-Unzip | PacBio (192 ×) | 243 | 224 | 1.121 | 511 | 105 | 511 | **105** | **0.00** | 82.28 | |
| | FALCON-Unzip, Pilon, PH | PacBio (192 ×) + PE | 232 | 222 | 1.109 | 512 | 105 | **512** | 105 | **0.00** | 82.49 | |
| H. sapiens | Platanus-allee | PE + 4 MP | 3,898 | 2,018 | 0.325 | 4 | 194,055 | 2 | 454,273 | 6.25 | 6.35 | |
| | | PE + 4 MP + PacBio (x20) | 5,684 | 5,406 | 0.872 | 306 | 5,294 | 19 | 78,213 | 6.72 | 59.71 | |
| | | PE + 4 MP + 10X | 4,918 | 4,025 | 0.649 | 59 | 23,612 | 13 | 118,218 | 2.64 | 39.15 | |
| | | PE + 4 MP + PacBio (20 ×) + 10X | 5,673 | 5,460 | **0.881** | 658 | 2,584 | 23 | 71,898 | 3.53 | 68.33 | |
| | FALCON-Unzip | PacBio (77 ×) | 4,721 | 3,691 | 0.595 | 109 | 13,508 | 109 | 13,508 | **0.00** | 30.36 | |
| | FALCON-Unzip, Pilon, PH | PacBio (77×) + PE | 4,851 | 3,783 | 0.610 | 124 | 11,817 | **124** | **11,817** | **0.00** | 30.98 | |
| | Supernova | 10X | 5,405 | 5,028 | 0.811 | 2,489 | 675 | 124 | 13,651 | 1.38 | 72.38 | |
| | Mostovoy et al. 2016 | PE + 1 MP + 10X + Bionano | 5,535 | 5,353 | 0.863 | **3,998** | **423** | 9 | 184,955 | 8.25 | **75.10** | |

Statistics were calculated for phased blocks whose length ≥ 500 bp. A bold value indicates the best one for each species. Bubbles are phased heterozygous regions. Genome sizes and heterozygosities were estimated based on the k-mer frequency information of PEs and GenomeScope[26]. Bubble-total/genome-size, NG50s and LG50s were calculated based on the estimated diploid genome sizes (P. polytes, 480 Mbp; B. japonicum, 780 Mbp; C. elegans, 200 Mbp; H. sapiens 6.2 Gbp). Estimated heterozygosities are shown in Supplementary Table 5. BUSCO[27] (version 3.0.2) was used to estimate the rate of the phased single-copy genes for P. polytes, B. japonicum, C. elegans and H. sapiens with the endopterygota set (2442 orthologs), the metazoa set (978 orthologs), the nematoda set (982 orthologs) and the euarchotoglires set (6192 orthologs), respectively

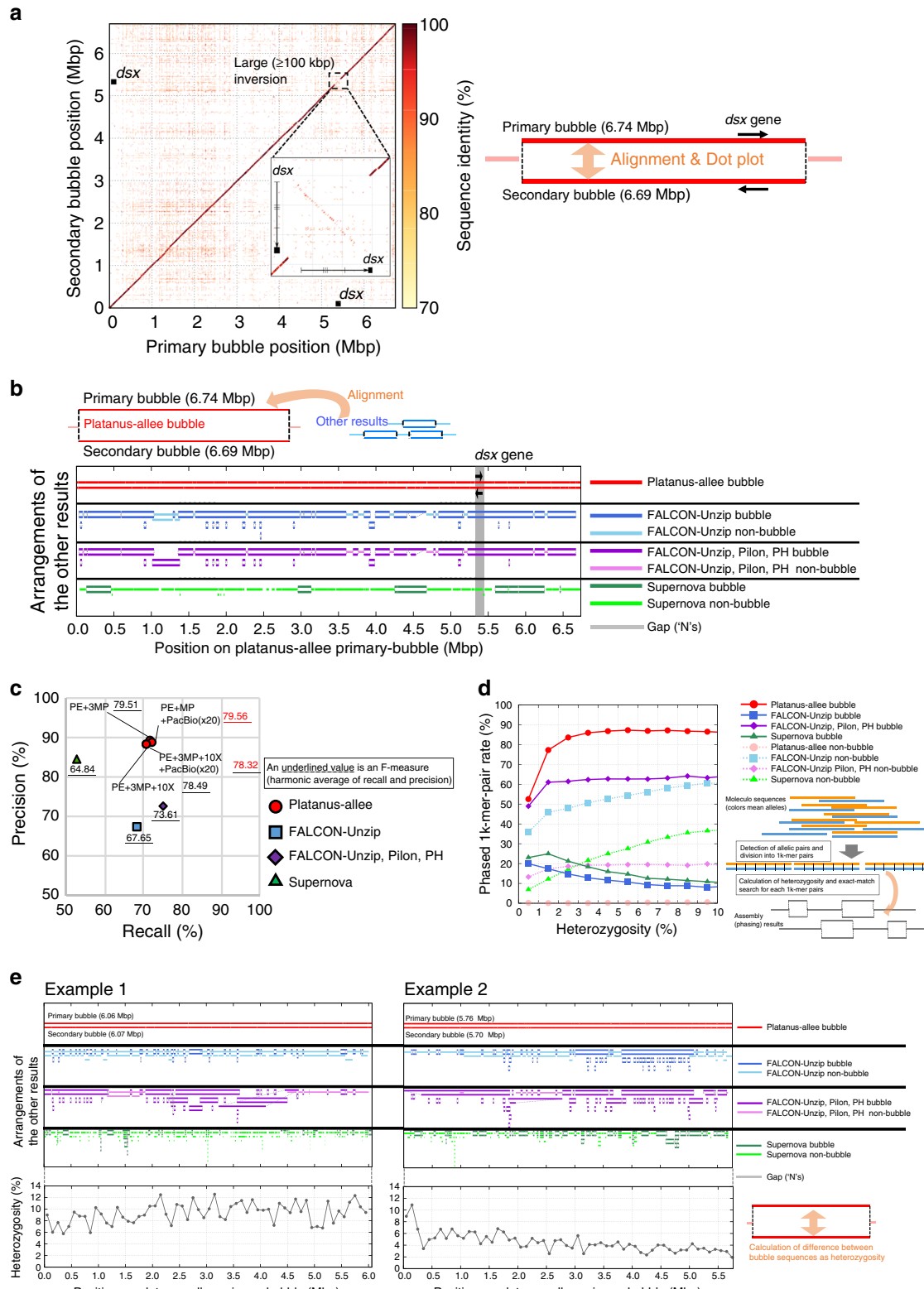

**Fig. 2** Benchmarking for *P. polytes* and *B. japonicum*. **a** Alignment between the *P. polytes* bubbles mapped to the *dsx*-gene locus. Dot plot was obtained using Nucmer and a modified version of Mummerplot in Mummer package[57]. **b** Alignment between Platanus-allee bubble and the results obtained using other methods for the *dsx*-gene locus. **c** Precision-recall evaluation of *B. japonicum* based on Molecule synthetic long reads. Underlined numbers indicate F-measures (harmonic mean of recall and precision). **d** Relation between phasing performances and heterozygosity of amphioxus. **e** Examples of highly divergent Platanus-allee bubbles obtained in the amphioxus analysis. Heterozygosities between bubble sequences were calculated based on the sequence difference (1 − (number of matches/alignment-length)) for the mapped 1k-mer pairs obtained from the Molecule data using 100 kbp-windows. Alignments ("Arrangements of other results") in **b**, **e** were performed using Minimap2[20] (see "Methods" section)

**Table 2 Consensus sequence statistics**

| Species | Assembler | Input data | Total (Mbp) | Total / geome-size | Scaffold NG50 (kbp) | Scaffold LG50 (#) | Contig NG50 (kbp) | Contig LG50 (#) | % gaps | BUSCO single complete (%) |
|---|---|---|---|---|---|---|---|---|---|---|
| *P. polytes* | Platanus-allee | PE + 4 MP | 248 | 1.032 | **8,204** | **10** | 157 | 460 | 3.22 | 95.95 |
| | | PE + 4 MP + PacBio(20 ×) | 247 | 1.028 | 7,784 | **10** | 164 | 438 | 3.18 | 96.19 |
| | | PE + 4 MP + 10X | 248 | 1.031 | 6,621 | 13 | 138 | 516 | 3.08 | 96.40 |
| | | PE + 4 MP + PacBio(20 ×) + 10X | 247 | 1.029 | 7,845 | 11 | 145 | 482 | 3.02 | 96.44 |
| | Platanus (v1.2.4) | PE + 4 MP | 232 | 0.968 | 5,995 | 15 | 159 | 438 | 1.37 | **96.52** |
| | FALCON-Unzip | PacBio(99 ×) | 261 | 1.088 | 5,196 | 18 | 5,196 | **18** | **0.00** | 89.35 |
| | FALCON-Unzip, Pilon, PH | PacBio(99 ×) + PE | 242 | **1.008** | 5,199 | 18 | **5,199** | **18** | **0.00** | 92.47 |
| | Supernova | 10X | 257 | 1.070 | 329 | 201 | 98 | 638 | 1.75 | 88.00 |
| *B. japonicum* | Platanus-allee | PE + 3 MP | 384 | 0.984 | 5,336 | 22 | 48 | 2,149 | 4.40 | 94.07 |
| | | PE + 3 MP + PacBio(20 ×) | 389 | **0.997** | 7,304 | **16** | 49 | 2,120 | 5.12 | **94.58** |
| | | PE + 3 MP + 10X | 392 | 1.004 | 4,914 | 21 | 34 | 3,104 | 4.52 | 93.87 |
| | | PE + 3 MP + PacBio(20 ×) + 10X | 397 | 1.017 | 5,413 | 17 | 35 | 3,030 | 5.15 | 94.07 |
| | Platanus (v1.2.4) | PE + MP | 488 | 1.250 | 239 | 414 | 10 | 10,393 | 13.89 | 65.75 |
| | FALCON-Unzip | PacBio(156 ×) | 707 | 1.812 | 4,301 | 32 | **4301** | **32** | **0.00** | 27.91 |
| | FALCON-Unzip, Pilon, PH | PE + PacBio (156 ×) | 406 | 1.042 | 3,259 | 39 | 3,259 | 39 | **0.00** | 84.97 |
| | Supernova | 10X | 662 | 1.697 | 45 | 2271 | 23 | 5,554 | 2.86 | 54.09 |
| *C. elegans* | Platanus-allee | PE + 3 MP | 106 | 1.058 | 2,388 | 14 | 63 | 458 | 4.76 | 95.52 |
| | | PE + 3 MP + PacBio(20 ×) | 107 | 1.065 | **3,316** | **12** | 64 | 456 | 5.13 | 95.11 |
| | Platanus (v1.2.4) | PE + 3 MP | 102 | **1.016** | 1,848 | 19 | 71 | 364 | 2.30 | **96.64** |
| | FALCON-Unzip | PacBio(192 ×) | 109 | 1.093 | 2,064 | 17 | **2,064** | **17** | **0.00** | 93.18 |
| | FALCON-Unzip, Pilon, PH | PacBio(192 ×) + PE | 103 | 1.029 | 2,064 | 17 | 2,064 | **17** | **0.00** | 94.60 |
| *H. sapiens* | Platanus-allee | PE + 4 MP | 2,894 | 0.934 | 3,917 | 230 | 23 | 34,650 | 4.08 | 88.82 |
| | | PE + 4 MP + PacBio(x20) | 2,995 | 0.966 | 3,717 | 250 | 22 | 36,365 | 6.76 | 86.45 |
| | | PE + 4 MP + 10X | 2,914 | 0.940 | 3,050 | 305 | 25 | 33,967 | 2.46 | 89.34 |
| | | PE + 4 MP + PacBio(20 ×) + 10X | 2,954 | **0.953** | 3,730 | 252 | 24 | 34,583 | 3.66 | 88.66 |
| | Platanus (v1.2.4) | PE + 4 MP | 2,706 | 0.873 | 13,149 | 65 | 24 | 33,902 | 1.62 | **90.39** |
| | FALCON-Unzip | PacBio(77 ×) | 2,788 | 0.899 | 8,670 | 98 | **8,670** | **98** | **0.00** | 87.97 |
| | FALCON-Unzip, Pilon, PH | PacBio(77 ×) + PE | 2,791 | 0.900 | 8,669 | 98 | **8,669** | **98** | **0.00** | 88.97 |
| | Supernova | 10X | 2,942 | 0.949 | 30,823 | **28** | 147 | 5,954 | 1.42 | 90.33 |
| | Mostovoy et al. 2016 | PE + 1 MP + 10X + Bionano | 2,857 | 0.922 | **30,830** | 34 | 9 | 91,880 | 10.22 | 88.76 |

Statistics were calculated for consensus (pseudo-haploid) sequences whose length ≥ 500 bp. *P. polytes*, *B. japonicum*, *C. elegans* and *H. sapiens* correspond to a butterfly, am amphioxus, a worm and the human (NA12878), respectively. A bold value indicates the best one for each species. Genome sizes were estimated based on the *k*-mer frequency information of PEs and GenomeScope[26]. Total/ genome-size, NG50s, and LG50s were calculated based on the estimated haploid genome sizes (*P. polytes*, 240 Mbp; *B. japonicum*, 390 Mbp; *C. elegans*, 100 Mbp; *H. sapiens* 3.1 Gbp). BUSCO was used to estimate the rate of the non-redundantly constructed single-copy genes in a similar manner for the phased blocks (Table 1). The formats of the results from FALCON-Unzip and Supernova were "primary-contigs" and "pseudohap," respectively

whether it matched the bubbles and calculated the heterozygosity (number of match-sites/alignment-length). Finally, we observed the relationship between heterozygosity and phasing performances (Fig. 2d). Remarkably, only Platanus-allee maintained the high rate of phased 1k-mer pairs in bubbles for high heterozygosity, though Pilon and PH recovered that of FALCON-Unzip, suggesting that only the Platanus-allee tool can be efficiently used for HDR analysis. Examples of long and highly divergent bubbles obtained using Platanus-allee are presented in Fig. 2e, whereas the results obtained using other tools were fragmented and rich in redundant non-bubble sequences, not suitable for downstream analysis.

Even as a consensus scaffold, Platanus-allee exhibited larger contiguities (scaffold-NG50) for the aforementioned heterozygous samples than the other tools (Table 2). Especially for the amphioxus, only Platanus-allee yielded the sequences that had total sizes similar to the estimated genome size (390 Mbp), the large scaffold-NG50s of consensus sequences (maximum, 7.3 Mbp) and the high BUSCO single-completeness (>95%), while the other tools may have generated redundant and/or fragmented

sequences. These results were consistent with the high performances of Platanus-allee, in terms of phasing, and performing the phasing procedure first will be effective for obtaining useful consensus genomes from highly heterozygous organisms.

**Benchmarking for model organism data.** To assess the phasing performances comprehensively, using the reference genomes, we generated synthetic diploid data from the *Caenorhabditis elegans* strains N2[29] and CB4856[30], whose genomic sequences have been completely determined with almost no gaps (genome size, 100 Mbp). These two strains are nearly completely homozygous, and we individually sequenced them and used the same amount of data in silico (Supplementary Table 3). The single nucleotide variant (SNV) density between the two genomes was reported as 0.33%, and the HDRs were shown to represent 2–16% of the genome[30]. Therefore, we considered these datasets suitable for analysis.

As a Platanus-allee competitor, we executed FALCON-Unzip because it exhibited better performance than Supernova for most indicators during previous analyses. We tested the coverage depths of PacBio from 80 × to 192 × for FALCON-Unzip and used 20 ×

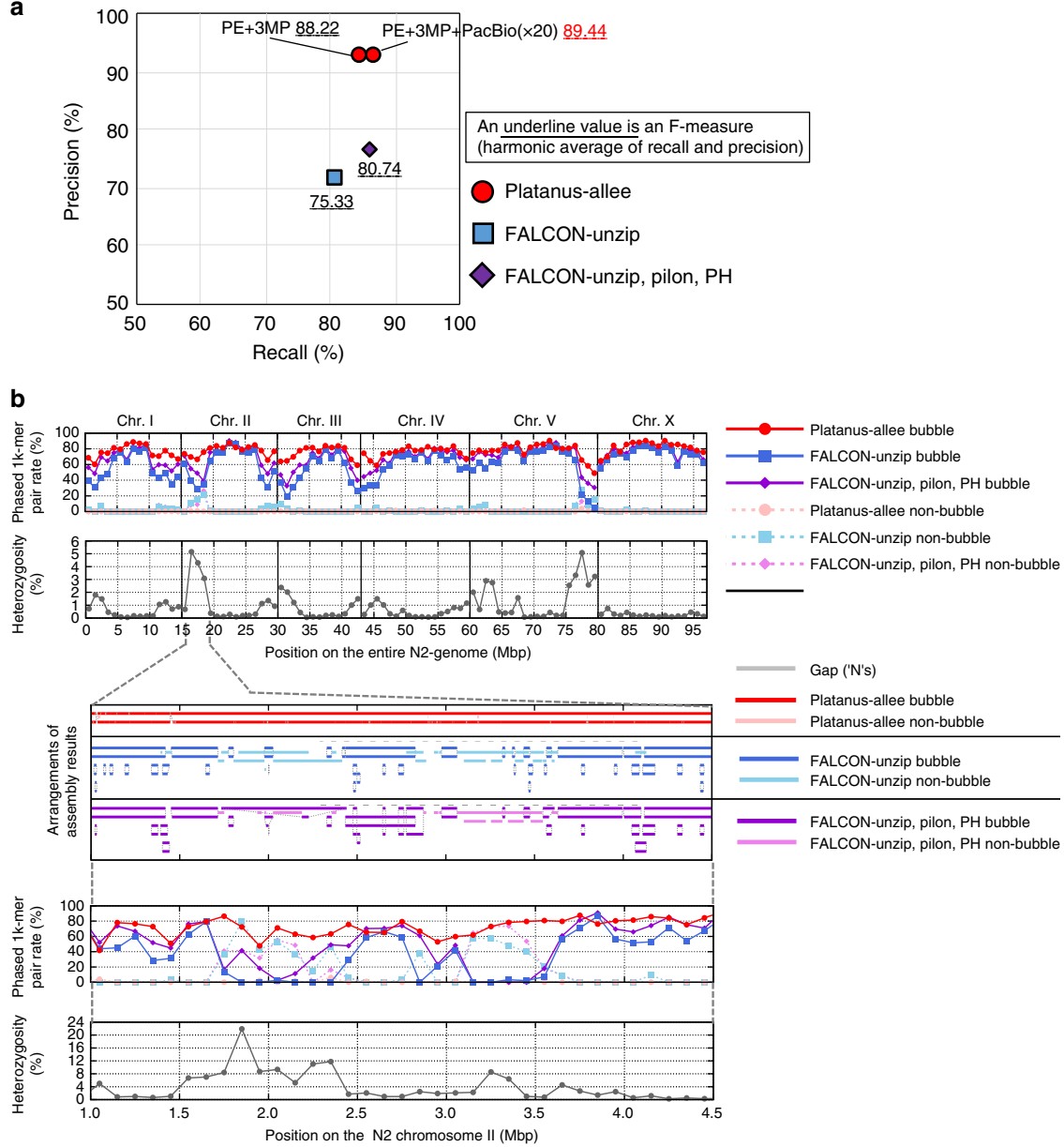

**Fig. 3** Benchmarking for the synthetic diploid sample of *C. elegans*. **a** Precision-recall evaluation for the synthetic diploid *C. elegans* based on the reference genomes. Underlined numbers indicate F-measures. **b** Phasing performances and heterozygosity for the entire genome and a highly divergent region of the synthetic diploid *C. elegans*. Phased 1k-mer pair rate and heterozygosity in the entire genome (above two graphs) and the highly divergent chromosome II end (bottom two graphs) were calculated using 1 Mbp- and 100 kbp-windows, respectively. Alignments ("Arrangements of the other results") were performed using Minimap2[20] (see "Methods" section)

for Platanus-allee. The recall and precision were determined using an approach similar to that used when analyzing amphioxus, but reference genomes were used instead of the Moleculo contigs. For the contiguities of the phased blocks, Platanus-allee results exceeded the quality of the best result obtained with FALCON-Unzip (192 ×) when three MPs were used (Table 1; Fig. 3a; Supplementary Table 7, 12). Similarly, for recall and precision, Platanus-allee produced larger values than FALCON-Unzip when inputting at least one MP. Remarkably, Platanus-allee with MPs invariably showed higher precision (differences > 10%) than FALCON-Unzip even with Pilon and PH.

To investigate the correlation between heterozygosity and phasing performance, we initially aligned the reference sequences of the two strains for each chromosome using a previously

described method[30], and heterozygosity (SNV density) was calculated for each 1 Mbp-window (see "Methods" section). The 1k-mer pairs from the alignment between reference genomes were then identified, and the rate of exactly phased 1k-mer pairs was calculated for each 1 Mbp-window as an indicator of phasing performance. The results (Fig. 3b) showed that the heterozygosities tend to be higher near the ends of the chromosomes, as previously reported[30], and the coverage obtained using Platanus-allee frequently exceeded that obtained using FALCON-Unzip in these regions. The highly divergent region near the end of chromosome II is magnified in Fig. 3b, demonstrating that FALCON-Unzip could not phase the HDR as bubbles. The mean heterozygosity of 1 Mbp-windows where Platanus-allee had much larger phased 1k-mer rate (differences > 10%) than the other tools did was 1.27% ($n = 23$),

which was significantly higher than that of the other windows, 0.68% ($n = 74$) ($p = 0.0192$, two-sided Wilcoxon rank-sum test).

Structural accuracy was determined by counting switch-errors (i.e., errors of phasing) and mis-assemblies. We determined the inconsistency between the assembly and the reference based on the alignments of fixed-length fragments (for details, Supplementary Note 18). Platanus-allee with PE and MPs was shown to lead to a lower rate of switch-errors + mis-assemblies compared to those obtained using FALCON-Unzip with 192× data (Supplementary Table 13).

Additionally, we confirmed that the consensus scaffolds of Platanus-allee indicated a relatively high contiguity, completeness, and accuracy (Table 2; Supplementary Table 14) even for the *C. elegans* data.

Finally, we benchmarked the tools using the human sample NA12878 (female, cell line GM12878). The input PE, MP (2 kbp), PacBio, and 10X data were previously generated and reported[15,16,23]. We additionally sequenced three MPs (5–15 kbp) (Supplementary Table 4). To the best of our knowledge, the de novo assembly-based phasing results with the largest N50 was previously reported in Mostovoy et al. 2016[31] utilizing PEs, 10X, and Bionano genome-map data, and we used this phased block set for comparison. For the calculation of precision and recall, Moleculo data of the 1000 Genomes Project[32] were used.

In striking contrast to highly heterozygous organisms, the contiguities of Platanus-allee and FALCON-Unzip in this analysis were shown to be inferior to the other tools probably due to the shortage of heterozygous sites used for phasing, and the scaffold-NG50 of phased blocks substantially increased with PacBio or 10X data (Table 1; Supplementary Table 7). Note that this tendency was common to the consensus sequences (Table 2; Supplementary Table 8). Scaffold-NG50s of phased blocks of Platanus-allee with PacBio were shown to be larger than the reported value (145 kbp) of pipeline[16] that combined the PacBio-assembly of FALCON and the mapping of Illumina reads, and in some cases, the value obtained with Platanus-allee surpassed that of the phasing method based on fosmid-pooling (484 kbp)[33] (target individual was not NA12878). The recall performance of Platanus-allee with PacBio or 10X exceeded that of the Mostovoy et al. 2016 pipeline[31], but was smaller than that of Supernova in all cases (Fig. 4a, Supplementary Table 15). In contrast, the precision of Platanus-allee was better than that of any other tool. Although the situation may be impractical, when all libraries including PacBio (20×) and 10X were input into Platanus-allee, its F-measure was the top value.

We also performed the evaluation using the phased variant set of NA12878, called "Platinum set", previously derived from the trio data[34]. Note that the variants were basically detected on the basis of mapping of short-reads and possibly HDRs were not reflected, but this evaluation was valuable as the one not depending on the Moleculo data. For Recalls and precisions measured based on fixed-length fragments from the haplotypes of the Platinum set (Supplementary Table 16), the trend was consistent with those for the Moleculo contigs. The details are described in Supplementary Note including the evaluation of switch-errors and mis-assemblies (Supplementary Table 17).

As previously shown with the amphioxus, the relation between heterozygosity and phasing performance (phased 1k-mer rate) was investigated based on the Moleculo contigs (Fig. 4b), and Platanus-allee showed a better performance in the HDR (heterozygosity ≥ 3%) analyses even when a human sample was used, although the fraction of HDRs was low compared to that observed in other organisms. One remarkable example of HDR was the major histocompatibility complex (MHC) region[35], which generally shows high haplotype divergence and functional importance. Platanus-allee successfully constructed this highly divergent bubble (~1.05

Mbp) covering the entire MHC class II region (Fig. 4c). Although Supernova also succeeded in the construction of a long bubble (~6.95 Mbp) covering the MHC region, the HDR between the *HLA-DRA* and *HLA-DRB1* loci was represented as a large gap (~90 kbp) and short sequences. In this region, FALCON-Unzip indicated the divisions of phased blocks and Mostovoy et al. 2016 could not construct the large range of the region (>300 kbp), illustrating the advantage of using Platanus-allee for HDR analysis. Notably, *HLA-DRB3* gene was found (identity ≥ 95% and alignment-coverage ≥ 50%) in one of the haplotype Platanus-allee constructed (Fig. 4d), which was concordant with the report based on the trio data and PacBio reads[13].

To evaluate the accuracies of the phased blocks of the MHC region, using the previous typing results of six loci in the MHC region[36], we counted loci correctly represented in the phased blocks (for details, Supplementary Note 24). As a result, (Supplementary Table 18), Platanus-allee with MP and/or PacBio and FALCON-Unzip correctly typed all loci, and the number of exact-matches to database-sequences was the most for Platanus-allee, supporting that this tool has an advantage to phase the MHC region.

Intriguingly, 8 long bubbles were observed (primary + secondary length ≥ 100 kbp) that were not found in the reference genome (GRCh38.p10) or the Supernova results when Platanus-allee was used (for details, Methods and Supplementary Table 19). The longest one (lengths, 435 kbp and 382 kbp) was highly diverged and repeat-rich (Fig. 4e). Mapping of the Moleculo contigs and PacBio reads confirmed that this bubble was not an artifact or a contaminant sequence. (Supplementary Fig. 6). Most of the other observed bubbles absent from the reference genome were shown to be of similar nature (Supplementary Fig. 7), and some exhibited an extremely high repeat rate.

**Benchmarking for the other organisms using the public data.** To further investigate the versatility of Platanus-allee, we added the two plants (*A. thaliana* F1-hybrid[18] and *Prunus yedonensis* (cherry blossom)[37]) and the one mammal (*Pteropus alecto* (bat)[38]) samples, whose sequencing data are publicly available (Supplementary Table 20–22). Although the performances of Platanus-allee were inferior to those of FALCON-Unzip when MP was not available (*A. thaliana*, Supplementary Table 23–25), the other benchmarks confirm the effectiveness of Platanus-allee for high heterozygosity and MPs, demonstrating the high performances compared to the public assemblies (Supplementary Table 23; for details, Supplementary Note 26).

**Benchmarking for fully simulated heterozygous data.** Finally, to systematically investigate the effect of heterozygosity, we prepared data sets including all types of libraries fully simulated in silico using the multiple tools[39–41]. The target species was *C. elegans* (N2) and the configuration of the libraries was similar to that in the benchmark of synthetic diploid *C. elegans*, and the heterozygosities were simulated as 0.1–2% using pIRS[42] (for details, Supplementary Note 27).

The results are shown in Supplementary Table 26, 27. Although most NG50s of both phased blocks and consensus sequences outperform those of the actual data possibly unknown natures of actual ones, there are two important tendencies:

(1) The performance of Supernova considerably worsened for 2%-heterozygous data. This is consistent with the bad performances of this tool for *P. polytes* and *B. japonicum*, implying that it cannot handle high heterozygosity (>1%).

(2) Platanus-allee exceeded FALCON-Unzip even with Pilon and PH when heterozygosity ≥ 1% for most indicators. This supports the advantage of Platanus-allee for HDRs.

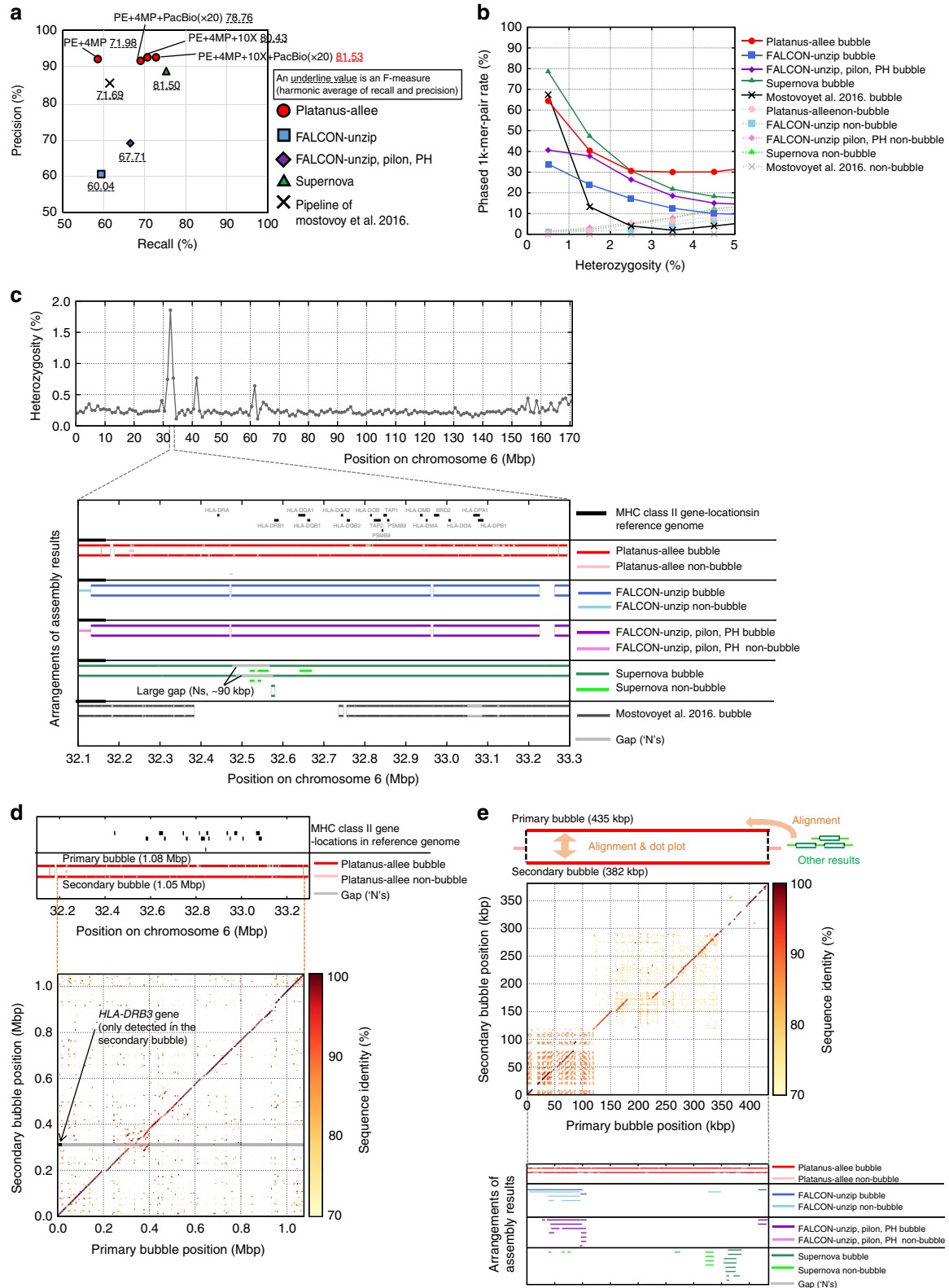

**Fig. 4** Benchmarking for *H. sapiens*. **a** Precision-recall evaluation of the human sample based on the Moleculo synthetic long reads. Underlined numbers indicate F-measures. **b** Relation between phasing performances and heterozygosity using a human sample. **c** Phasing results for the human MHC class II region. Positions on chromosome 6 and gene annotation were based on human reference genome GRCh38.p10. Heterozygosity was calculated based on the sequence difference (1 − (number of matches/alignment-length)) for the mapped 1k-mer pairs from the Moleculo data using 1 Mbp-windows. **d** Dot plot between the bubble of Platanus-allee covering the MHC class II region. **e** Highly divergent and non-localized bubble obtained using human NA12878 data. Alignments ("Arrangements of the other results") in **c**, **e** were performed using Minimap2[20] (see "Methods" section). Dot plots in **d**, **e** were depicted using Nucmer and modified version of Mummerplot in Mummer package[57]

## Discussion

The ultimate goal for diploid eukaryotic species is to completely decode each homologous chromosome sequence, and a comprehensive survey of HDRs can be performed by comparing homologous pairs of chromosome sequences.

Platanus-allee was evaluated for determination of homologous chromosome sequences and it showed high performance especially for HDRs. In each case, the total size of its phased blocks was close to twice the size of each genome, and the consensus sequences were close to each genome size, while FALCON-Unzip and Supernova built too large or too small sizes. Considering these benchmarks, we conclude that the Platanus-allee approach has versatility and advantages in the analysis of non-model organisms that often have high levels of heterozygosity.

The algorithm of Supernova, which utilizes short-reads and bubbles in de Bruijn graphs, is partially similar to that of Platanus-allee. However, the performances of Supernova obviously declined for the highly heterozygous samples and HDRs (Table 1, Fig. 2d and so on), displaying the difference between these tools.

The range of real times to run Platanus-allee with 20 threads were 4.54–13.5 days for the human, while that was 6.44–8.45 h for *C. elengans* (Supplementary Table 9). Although this tool is not designed to make use of machine clusters, we could complete the assembly of human genomic reads composed of PE, MPs, 10X and PacBio within 2 weeks in a single machine, suggesting that Platanus-allee is applicable to multiple genome projects.

Through four benchmark analyses, while using Platanus-allee, Illumina MP libraries were shown to be effective for the analysis of highly heterozygous genomes, and at least one MP can substantially improve the performance of the approach even when combined with PacBio data. In contrast, PacBio and 10X data were more effective when analyzing a lowly heterozygous sample. This is associated with the concept of Platanus-allee that exactly separates the haplotype sequences using accurate reads, and MP is the practical choice to obtain long-range link information with low sequencing error rate (<1%). Although MP is not a state-of-the-art method in comparison to PacBio or 10X, it was recently employed for the phasing of polyploid plants[43,44] and de novo assembly of the human genomes[45] as well. The NG50s of phased blocks or consensus sequences in the latter case were comparable to those obtained with PacBio, suggesting that the usefulness of MP is not specific only to Platanus-allee. In contrast, combination of PE and PacBio without MP does not lead to a better result of Platanus-allee compared to that of FALCON-Unzip for many cases.

Platanus-allee generated many gaps compared to the other tools. Especially FALCON-Unzip constructs completely gap-free sequences as its algorithm design making use of long reads. Long-reads-based strategy will also cover extremely repeat-rich regions[46]. However, Platanus-allee indicated better performances for recalls and BUSCO-completeness in most cases. Importantly, precisions of Platanus-allee outperformed those of FALCON-Unzip even when Pilon was applied. We believe that the results of Platanus-allee have a certain value for practical uses.

Although describing the exact costs are difficult because of the rapid improvement of technology and dependencies of retail prices on distribution channels, the rough summaries are described below referring the previous comparison and reviews[47–49]. PE is the most cost-effective one and it is useful for preliminary survey, assembly and polishing. The sequencing cost of a typical configuration of MPs (3–4 libraries, ~40× for each) used in this study may be lower than that of a high-coverage (≥100×) PacBio library preferred from FALCON-Unzip, although the relation is possibly reversed for the samples with smaller genome sizes. For lowly heterozygous samples such as humans, 10X combined with Supernova might be the most cost-effective strategy if HDRs are out of interest (for details, Supplementary Note 30).

Lastly, we propose three possible applications of the de novo-assembled phased block. First, it can be used for the cataloging of variants, including structural ones. The effectiveness of de novo assembly for complex variants has been validated using human samples[15,45] and it is expected to be more important for highly heterozygous organisms such as sea squirt[5] and lancelet[3]. Furthermore, the phased blocks are helpful for the investigation of the allele-specific nature of diploid genomes, such as gene expression (RNA-Seq), DNA methylation (bisulfite-seq), protein binding (ChIP-seq) and chromosome conformation (Hi-C). For the methods based on sequencers, the possible targets are not limited to those listed above. Allele-specific gene expression and DNA methylation have been detected in the phased blocks of human sequences based on fosmid-pooling[33], and recent single-cell RNA-seq studies reported that 12–76% of genes may be expressed monoallelically[50,51], emphasizing the importance of this analysis. In the *H* locus of *P. polytes*, not only differences in gene expression but also a haplotype-specific non-coding gene, *U3X*, were observed[2]. Finally, we believe that certain fields, such as phylogenetics and population genetics, that regularly use genome comparison in their analyses will greatly benefit from the use of phasing. Phased blocks can be used to determine the origin of specific haplotypes, exemplified in the phylogenetic study of beer yeasts[52]. Other examples include the application of phasing results for high-resolution inference of population history[53] and the detection of selective pressure for each haplotype[54]. Overall, the accuracy and the comprehensiveness of the phased blocks generated using Platanus-allee are expected to facilitate a large variety of downstream analyses.

## Methods

**Sample collection**. One wild individual of *P. polytes* was collected in Ishigaki Island, Okinawa, Japan in 2016. No field work permit was required for sampling this species in this region. One wild individual of *B. japonicum* was collected Atsumi Peninshula, Aichi, Japan in 2012. The samples of *C. elegans* N2 and CB4856 strains were the laboratory stocks. The samples of *H. sapiens* was the cell line of GM12878, CEPH/UTAH Pedigree 1463, provided by Coriell Institute.

**Sequencing**. The Illumina paired-end and mate-pair libraries were prepared using TruSeq DNA PCR-Free LT Sample Prep Kit and Nextera Mate Pair Sample Prep Kit, respectively. The 10X linked-read libraries were prepared by the Chromium system (10X Genomics). The Moleculo library of *B. japonicum* was prepared using TruSeq Synthetic Long Read Kit. All Illumina sequencing runs, including 10X linked-reads and Moleculo, were performed using HiSeq 2500. PacBio sequencing runs were performed using RS II sequencer for *P. polytes* and *B. japonicum*, and Sequel for *C. elegans* N2 and CB4756 strains.

**Contig-assembly**. The contig-assembly module of Platanus-allee is derived from that of Platanus with some differences as follows:

(1) The initial *k*-mer coverage cutoff is set to (leftmost-local-minimum-coverage/2) rather than (leftmost-local-minimum-coverage). The leftmost-local-minimum-coverage is determined for an occurrence distribution of *k*-mers (default, $k = 32$) using a smoothing window-size of 1 rather than 7.

(2) In steps of increasing $k$ and reconstruction of the de Bruijn graph, a step size of $k$ ($k_{step}$) is set to 20 rather than 10.

(3) The function to correct contigs based on mapping of the reads ("Check of contigs using exact-match reads" in ref. [1]) is omitted.

(4) The bubble-removal function is omitted.

Steps (1) and (3) above are adopted to handle low-coverage heterozygous regions, which are expected to cover half of the homozygous regions.

**Construction of a (gapped) de Bruijn graph**. Similar to Platanus, Platanus-allee extracted sub-graphs in a de Bruijn graph (*k*-mers) without junctions, which have multiple edges for the same direction, and treated them as "straight nodes" (contigs). The nodes corresponding to junctions are "junction nodes". Input to the phasing step described in sections below is a de Bruijn graph ($k = k_{max}$, see ref. [1])

resulting from the contig-assembly module, which consists of both straight and junction nodes.

Straight nodes in the graph may contain gaps (Ns) after the first iteration round, which was referred to as a gapped de Bruijn graph in this study. Note that a pair of nodes are connected if they have $(k - 1)$-length overlap with each other.

**Mapping reads to a (gapped) de Bruijn graph.** For short reads (Illumina), the reads are mapped to the nodes of the graph in a similar manner as in Platanus based on exact and unique matches of fixed-length sequences. The differences from Platanus are as follows:

(1) Three fixed-lengths for mapping are applied, 32, 64, and 96, to increase sensitivity.

(2) If single read is mapped to multiple nodes, link information among those nodes are stored and possibly used in downstream steps. Note that Platanus only uses links between paired-reads and/or mate-pairs.

Long reads (PacBio or Nanopore) are mapped to the nodes using Minimap2[20] (version 2.0-r191 in this study) with default parameters except for the options for detailed alignments (-c) and multi-threading (-t user-specified-value). In this setting, the $k$-mer size for mapping is 15 (-k 15) and error-prone long reads can be handled. After execution of Minimap2, the resulting alignments are filtered based on the following conditions (1):

$$sequence - identity \geq 0.8 \text{ and } (alignment - length \geq 1000 \text{ or } alignment - coverage \geq 0.8) \quad (1)$$

Here, the sequence-identity and alignment-coverage are defined as follows (2, 3):

$$sequence - identity = (\# \text{ matched} - sites/alignment - length) \quad (2)$$

$$alignment - coverage = alignment - length/\min(query - length, target - length) \quad (3)$$

Next, the alignments are greedily selected according to the order of # match-sites, ensuring that each pair of selected alignments does not have overlaps $\geq k$ ($k$-mer size) in a query-sequence. For each pair of query and target sequences, the positions of local alignments are weighted by their lengths and the weighted average position is used as the final mapped position. If one query sequence (read) spans multiple target sequences (nodes), the gap size between nodes is determined from the read sequence, and the resulting link information is stored and used in downstream steps.

The mapping result of each library is used individually in ascending order of insert-sizes and finally all libraries are used simultaneously for each iteration (for details, Methods, Iteration of phasing).

**Mapping linked-reads to a (gapped) de Bruijn graph.** Linked-reads are paired-end reads containing short (~16 base pair, bp) "barcode" sequences generated from the 10X Genomics platform. Platanus-allee assumes that input linked-reads are formatted as FASTQ files in which barcode sequences are represented as "BX: Z:" tag information. These files can be generated by the "longranger basic" command of Long Ranger, which is found on the web page of 10X Genomics (https://support.10xgenomics.com/genome-exome/software/downloads/latest).

Linked-reads are mapped to the (gapped) de Bruijn graph in a similar manner for paired-ends (see "Methods" section, Mapping reads to a (gapped) de Bruijn graph), and the number of reads is counted for each node and barcode. To reduce the effect of mis-mapping, the counting results are discarded if the number of reads $\leq 3$.

**Detection of anchor bubbles for haplotype synteny-based correction.** For the initial de Bruijn graph, simple bubbles, which consist of two straight nodes, two junction nodes, and four edges ($(k - 1)$-length overlaps), are extracted to estimate the sequence coverage depth corresponding to non-repetitive regions in a genome. Here, similarly to Platanus, the sequence coverage depth of each node in a de Bruijn graph is estimated from the $k$-mer coverage depth. Most extracted simple bubbles are assumed to be heterozygous regions, and the length-weighted average value is calculated from the coverage depths of straight nodes inside the bubbles as the coverage depth of non-repetitive heterozygous regions, $c_{hetero}$. For each simple bubble in the initial de Bruijn graph, if the coverage depths of both straight nodes $\leq r_{upper\text{-}threshold} \times c_{hetero}$ ($r_{upper\text{-}threshold}$ is the constant value; default, 1.75), this value is used as an "anchor bubble" for haplotype synteny-based correction as described below.

**Untangling cross structures in the (gapped) de Bruijn graph.** By definition, a cross structure consists of five straight nodes and two junction nodes (Supplementary Fig. 2a). Note that the edges between nodes above are based on the de Bruijn graph, which reflects $(k - 1)$-length overlaps of sequences. Next, for simplification, one straight node and the flanking two junction nodes are merged into one node, named as the "center node", while the remaining four nodes are named as the "external nodes" (Supplementary Fig. 2b). Let $c_{hetero}$ be the estimated average coverage depth of heterozygous nodes (for details, see "Methods" section, Detection of anchor bubbles for haplotype synteny-based correction) and $r_{upper\text{-}threthold}$ be the

constant value to determine a threshold (default, 1.75). To phase non-repeat heterozygous regions, the untangling function targets cross structures that satisfy the following conditions (4, 5):

$$coverage - depth - of - the - center - node \leq 2\times r_{upper-threshold} \times c_{hetero} \quad (4)$$

and

$$\min(coverage - depths - of - the - four - external - nodes) \leq r_{upper-threshold} \times c_{hetero} \quad (5)$$

Platanus-allee determines whether a cross structure is untangled as one of two possible solutions (depicted as parallel and cross in Supplementary Fig. 2c), for which determination methods are available as follows:

(1) According to the number of links

For each solution (parallel or cross), the number of supporting links between external nodes is counted for paired-ends, mate-pairs, short-reads, and/or long-reads (see "Methods" section, Mapping all reads to a (gapped) de Bruijn graph). Let $n_{alt\text{-}link}$ be the number of the links supporting the alternative solution. One solution is adopted as the result of untangling if the number of the links of the solution $\geq 4 \times n_{alt\text{-}link}$.

(2) According to the number of match-sites in alignments of long-reads

This function is designed for long-reads to select one solution with a higher alignment-score against alternative one for each cross structure. Similar to (1), for each solution, the number of match-sites (calculated by the Minimap2 aligner) in supporting links between external nodes is summed. Let $m_{alt}$ be the number of match-sites supporting the alternative solution. Let $n_{alt\text{-}match}$ be the sum of the match-sites in supporting links for alternative solution. One solution is adopted as the result of untangling if the number of match-sites for the solution is $\geq 4 \times n_{alt\text{-}match}$.

(3) According to barcode-information of linked-reads (Supplementary Fig. 3)

If the length of a center node is $\geq 200$ kbp, this function is not executed because of the limitation of lengths of DNA fragments for linked-reads. For common barcode sets between external nodes, the number of reads is summed and assigned for each resolution (for details regarding counting reads for barcodes, see "Methods" section, Mapping linked-reads to a (gapped) de Bruijn graph). Let $n_{alt\text{-}barcode}$ be the number of reads comprising the common barcode set in alternative solution. One solution is adopted as the result of untangling if the number of reads comprising the common barcode set the solution $\geq 4 \times n_{alt\text{-}barcode}$.

Each untangling procedure above is iteratively executed. The iteration is ended if the number of iteration reaches five or number of untangled subgraphs is zero.

**Construction of the scaffold graph.** The resulting sequences of untangling of the de Bruijn graph (see "Methods" section, Untangling cross structures in the (gapped) de Bruijn graph) is input into the scaffolding module, which is derived from that of Platanus. The extension is that Platanus-allee can accept data from long-reads and handle multiple libraries simultaneously to construct a scaffold graph.

**Untangling cross structures in the scaffold graph.** For a scaffold graph, cross structures are detected and solved in a similar manner as in the untangling function for de Bruijn graphs (see "Methods" section, Untangling cross structures in the (gapped) de Bruijn graph). The difference is that edges between the center and external nodes are not necessarily derived from $(k - 1)$-length overlaps of sequences, but can be derived from link-information of read mapping. Short nodes are excluded from the scaffold graph according to the procedures of scaffold-graph construction; however, nodes containing anchor bubbles (i.e., candidates of heterozygous nodes) are not excluded.

**Haplotype synteny-based correction.** The summary of characteristic procedures associated with this step are as follows:

(i) Anchor bubble detection

Bubble structures in the initial de Bruijn graph are used as the anchors to associate homologous regions, assuming each HDR is flanked by the regions with the heterozygosity low enough to form bubbles. The bubbles derived from the repetitive sequences are excluded, using coverage depth information.

(ii) Alignment of homologous haplotype sequences

Haplotype sequences extended through untangling and scaffolding are aligned according to anchor bubbles, and homologous regions are determined. Omitting global sequence alignments, local structural differences between homologous pairs are permitted.

This function is applied to a (gapped) de Bruijn graph or a scaffold graph, and the schematic model is shown in Supplementary Fig. 3a. Each node of these graphs is a sequence consisting of nodes of the initial de Bruijn graph (contigs) and gaps (Ns) and has a numerical identifier (scaffold-ID). The procedures to correct one node, which consists of $n$ contigs, are as follows:

(1) Anchor bubble information (see "Methods" section, Detection of anchor bubbles for haplotype synteny-based correction) to find the homologous node are assigned to the contigs. Let $b_1, b_2, ..., b_n$ be the numerical sequences. If the $i$-th contig ($i \in \{1, 2, ..., n\}$) is an anchor bubble, $b_i$ is the scaffold-ID of the node to which the counterpart of $i$th contig belongs; otherwise $b_i = 0$.

(2) $B_{max}$, the scaffold-ID of the homologous counterpart of most anchor bubbles, is determined. Let $l_1, l_2, ..., l_n$ be the lengths of the $n$ contigs. $L_{sum}(x)$ is the sum of all $l_i$ such that $b_i = x$, and

$$B_{max} = \arg\max_{x \in N} L_{sum}(x) \qquad (6)$$

Note that $x$ is a scaffold-ID and non-zero integer.

(3) $d_{min}$ and $d_{max}$ are the minimum and maximum $i$ such that $b_i = B_{max}$, respectively, and the candidate position to be divided for correction. Note that it is assumed that the region from the $d_{min}$-th to $d_{max}$-th contig retains synteny to the homologous counterpart.

(4) Whether the region from the $d_{min}$-th to $d_{max}$-th contig contains many anchor bubbles whose counter-part-scaffold-IDs are not $B_{max}$. If so, $d_{min}$ and/or $d_{max}$ are corrected. For $d_{min}$, the division score is calculated as follows:

$$S(x) = \begin{cases} S(x-1) \text{ if } b_x = 0 \\ S(x-1) - l_x \text{ if } b_x = B_{max} \ x \in \{d_{min}, d_{min}+1, ..., d_{max}\} \\ S(x-1) + l_x \text{ otherwise} \end{cases} \qquad (7)$$

If positions at which $S(x) > 0$ exist, the position at which $S(x)$ is the maximum for $x$ is the new $d_{min}$. This correction procedure is applied for $d_{max}$ in a similar manner.

(5) The node is divided and the region from the $d_{min}$-th to $d_{max}$-th contig becomes the new node.

(6) Platanus-allee assumes that both ends of a homologous sequence-pair in a bubble correspond to each other as anchor bubbles. The schematic model of this step is shown in Supplementary Fig. 3b. Here, the $i$-th contig in a node is referred to as the "edge contig" if $i = 0$ or $i = n$; otherwise, it is called the "internal contig". A node is divided at the position of internal contigs if its counterparts are the edge contigs of other nodes.

The procedures described above are iteratively executed until the number of corrections (divisions) is zero. After iteration, nodes containing anchor bubbles are assumed as heterozygous bubble regions and are paired according to the anchor bubbles. The pairing information is used in the downstream steps.

**Gap-closing**. The gap-closing module, which maps paired-ends (mate-pairs) and assembles reads assigned to each gap, is derived from that of Platanus. This module is executed for results from the phasing module, and the steps through phasing and gap-closing are iterated twice (default) to output the final phased blocks (Fig. 1a). Note that mapping of reads is based on unique-hits (i.e., reads that are mapped to multiple regions are not used) and it is expected that haplotype phase information is retained without chimeric sequences.

**Construction of consensus scaffolds**. On the basis of the results of the phasing module, primary-bubbles and non-bubble sequences are input into the consensus-scaffolding module. First, $(k − 1)$-length overlaps of input sequences are detected ($k$ is the mer-length of the input de Bruijn graph for the phasing module). Each pair of input sequences is connected if the pair overlaps with each other without overlapping other sequences. Next, the connected sequences are input into the scaffolding procedure applied in the phasing module (Supplementary Note 1), and the resulting sequences are referred to as consensus scaffolds.

**Constructing heterozygous 1k-mer pairs from Moleculo data**. The heterozygous 1k-mer pairs were used to benchmark the phasing performance of the tools for amphioxus and human (NA12878) samples. The procedures to construct 1k-mer pairs are as follows:

(1) Bases in the Moleculo contigs with quality values ≤ 40 were masked as "N". Quality values were obtained from the FASTQ files.

(2) Masked contigs were aligned to themselves using Minimap2 (version 2.9-r720) with the options of "-c -k 19 -p 0 -Xc --cs=long". Note that the version of Minimap2 differs from those in the other sections, considering the usability of file-format conversions for downstream steps.

(3) An alignment is referred to as a near-exact-match alignment if edit-distance ≤ # N-bases-in-both-aligned-regions.

The information for edit distances and N bases in aligned regions were obtained from the values of "NM:i" and "cs:Z" in the PAF files, respectively. To exclude self-alignments and alignments between the same haplotypes, near-exact-match alignments were discarded.

(4) For each Moleculo contig, the best hit, which showed the largest number of match-sites, was extracted from the result of step (3).

(5) From the results of (4), alignments with a sequence-identity ≥ 0.8 and alignment-length ≥ 500 were extracted. The resulting alignments were assumed to be between homologous regions from different haplotypes.

(6) The result of step (5), represented in PAF file format, was converted to MAF file format. The conversion was conducted using "paftools.js", which is an attachment script of Minimap2 (version 2.9-r720).

(7) Each alignment in the MAF file was divided into non-overlapping blocks in which the shorter sequence contained 1k bases ('-'s were not counted). The blocks containing 'N' or no differences (i.e., homozygous) were discarded. The pairs of sequences in the resulting blocks were used as 1k-mer pairs as described below.

(8) To further reduce the effect of sequencing errors, the 1k-mer pairs were aligned to all Moleculo contigs and the number of exact-matches was calculated. The 1k-mer pairs in which both sequences contained at least two exact-matches, were extracted and used as benchmarks for phasing performances of the tools.

**Whole-genome alignment between the C. elegans strains**. The reference genomes of the *C. elegans* N2 and CB4856 strains were aligned using a similar procedure to that described previously for the CB4856 strain genome[30]. The specific procedures were as follows:

(1) For each nuclear chromosome (I–V and X), the reference sequences of the two strains were aligned using the LASTZ aligner[55] (version 1.03.66) with the options of "-gap=400,30 -hspthresh=3000 -gappedthresh=3000 -masking= 0 -format=LAV".

(2) For chaining of the alignments, the algorithm of ChainNet[56] is applied. Specifically, commands in the kentUtils toolkit (version v302.1; URL, https://github.com/ENCODE-DCC/kentUtils) were sequentially executed for the LASTZ-alignments (LAV file) as follows: lavToPsl, axtChain (-linearGap=loose -psl), chainMergeSort, chainPreNet, chainNet and netToAxt. The file-formant of the result was AXT.

(3) Alignments between regions from different strands were excluded and the sequence difference, (1 − (# match-sites / # aligned-bases)), was calculated for each 1 Mbp-windows on the N2-strain reference chromosome. The sequence differences were assumed to reflect the heterozygosity of synthetic diploid data (Fig. 3b).

(4) Each alignment in the AXT file was divided into non-overlapping blocks in which shorter sequence contains 1k bases ('-'s were not counted). The blocks containing 'N' or no differences were discarded. Pairs of sequences in the resulting blocks were used as 1k-mer pairs for the benchmarks of phasing performances of the tools in a similar manner to Moleculo contigs (see "Methods" section, Construction of heterozygous 1k-mer pairs from Moleculo contigs).

**Detecting sequences absent from the human reference genome**. For the phased heterozygous blocks (bubbles) constructed from Illumina and PacBio data using Platanus-allee, we searched those not present in the reference genome (GRCh38.p10) based on the alinements of 1k-mers in the phased blocks. The procedures were as follows:

(1) The phased blocks of Platanus-allee were divided into non-overlapping and fixed-length (1 kbp) fragments. Those containing gaps ('N's) were discarded.

(2) The fragments were aligned to the reference genome using Minimap2 (version 2.0-r191) with the options of "-c -k 19 -p 1".

(3) Alignments with an alignment-length < 500 or sequence-identity < 0.9 were extracted and treated as reference-unaligned.

(4) Reference-unaligned fragments were aligned to the Moleculo contigs sets using Minimap2 with the same options as in step (2). The fragments without an exact-match to the Moleculo contigs were discarded. This step confirms that reference-unaligned fragments were not caused by mis-assemblies.

(5) The rate of reference-unaligned fragments confirmed by the Moleculo contigs was calculated for each pair of bubbles.

(6) The aligned-fragments-rates against the phased blocks of the other tools (FALCON-Unzip (raw and with Pilon, PH), Supernova, Mostovoy et al. 2016 pipeline[31]) were calculated in the same manner. Bubbles whose lengths (primary + secondary bubbles) ≥ 100 kbp and unaligned-fragments-rate ≥ 0.25 for the reference genome and phased blocks of the other tools were extracted (Supplementary Table 19).

## Data availability

The whole-genome sequencing data generated in this study have been deposited under BioProject PRJDB7193. All other data are contained within the article and its supplementary information or upon reasonable request from the corresponding author.

## Code availability

The source code of Platanus-allee (version 2.0.2) are freely available under GNU General Public License (GNU GPL) version 3 and can be downloaded from http://platanus.bio.titech.ac.jp/platanus2.

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

## Acknowledgements

We would like to thank the staff of Comparative Genomics Laboratory at NIG and Itoh Laboratory at Tokyo Tech for supporting genome sequencing. The experiment using TruSeq Synthetic Long Read Kit (Moleculo) is supported by Illumina inc. (San Diego, CA). This work was supported by MEXT KAKENHI Grant Number JP16H06279, JP16H04719, JP15H05979 and AMED Grant Number JP16gm6010003.

## Author contributions

T.I. designed and supervised this study. Y.M., H.K., A.F., K.K., Y.K., and A.T. collected and prepared the DNA samples. A.T. performed genomic DNA sequencing. R.K. developed the software. R.K., D.Y. and M.O. performed the benchmarks. R.K. and T.I wrote the manuscript.

## Additional information

**Competing interests:** The authors declare no competing interests.

