## [peer review file · Nature Communications]

Reviewer #1 (Remarks to the Author):

The authors develop a new method to assemble each haplotype in highly divergent regions of genomes. As far as I know, it uniquely takes advantage of a variety of technologies in a De Bruijn graph to help solve this problem. They find that in highly heterozygous organisms and in highly divergent regions like the human MHC, their method performs better than existing methods. Since it is not designed for low heterozygosity regions, it does not perform as well in much of the rest of the human genome as expected, and the authors clearly state this limitation. I think this will be a valuable addition to the literature after addressing the comments below.

On page 4 the authors state “The FALCON-Unzip15, the phasing tool for the single-molecule DNA sequencer, was used to construct long phased blocks for heterozygous plants and fungi, but the heterozygosities were not high.”

- Could the authors give numbers to “not high” and state what they mean by “high”?

The authors state “Consequently, the development of a phasing tool able to handle these HDRs is one of the most urgent requirements for the comprehensive survey of diploid genome diversity.”

- This may be overly superlative. I recommend “an important requirement” instead of “one of the most urgent requirements”

Do I understand correctly from the supplementary tables 1-2 that the authors used only 20x PacBio coverage for Falcon-unzip? If so, this may be partly why it does not perform as well since I recall higher coverages were used in their paper.

Did the authors use the recommended coverage for Supernova (I think older versions wanted ~50-60x? Which version of Supernova was used?

I could not find where the authors made available the new data generated for this study. Could the authors clearly state this?

I could not find the code for Platanus-allee. For this to have the highest impact, the code should be made publicly available with documentation prior to publication.

Reviewer #2 (Remarks to the Author):

The authors of the paper "Platanus-allele: a de novo haplotype assembler enabling a comprehensive access to divergent heterozygous regions" describe a new de novo assembler that is focused on providing diploid scaffold results. This is indeed a very important step towards a better representation and more insights into diploid genomes. The authors describe the multiple steps needed to form these phased scaffolds and their improvements over the previous version Platanus. They compare Platanus-allele to the current established Falcon Unzip and Supernova from PacBio and 10x, respectively.

In the following I list my questions and concerns:

1. Your comparison is not totally fair. Falcon Unzip generates Contigs with Pacbio only and Supernova results in Scaffolds using 10x genomics reads only. Whereas Platanus-allele uses multiple technologies and even different libraries of illumina sequencing. For example, when you compare Platanus-allele using PE+ pacbio vs. Falcon Unzip you see that they both perform very similarly and in multiple cases Falcon Unzip outperforming Platanus (Table 1 and 2). Furthermore, you are not discussing the % of gabs or the total sum of gab bp in the assembly.
2. I am also concerned that the 1kmer match strategy for the molecular reads for the evaluation is biased against the pacbio data set. Everyone uses illumina reads to improve the sequence consensus quality of the contigs and I don't think you used that before requiring a 100% match.
3. I am also a bit surprised about the low performance of Supernova even after scaffolding. Or did you used the contigs only?
4. I am also missing a discussion or table about the runtimes of each program.
5. I would encourage you to discuss the cost vs. benefits for the different technology combinations Platanus-allele uses.
6. Have you tried to combine all technologies for Platanys ? So 10x + Pacbio + all illumina libraries?
7. What confuses me a bit is the question if Platanus is optimized only for these highly heterozygous crosses/ animals and thus its utility to more general genomes. Maybe you can comment on this a bit more in the discussion since the results are quite different between human and the eg. Butterfly. What would you expect in terms of plants, or other mammals?
8. I would also encourage you to include the comparison to Platanus the previous version of the de novo assembly software to highlight the improvements over this implementation.

Reviewer #3 (Remarks to the Author):

The authors present an extension of a previous assembler, Platanus, extending it to support more datatypes for haplotype resolution. It is a unique assembler in its support for multiple datatypes while most competing methods rely only on one. I was able to run the assembly on the author provided test dataset.

There are several issues I had with the current draft.

1. The tables present N50 stats but it is not clear if these are contig or scaffold statistics. Based on the stats, these appear to be scaffolds. However, FALCON-Unzip only produces contigs so this comparison is not fair. It is also not clear if the genome size is kept constant between different assemblies. The statistics using a consistent genome size and contigs should be presented for all assemblers.
2. Figure 1 panel a is not clear to me. The figure shows the reads being binned by maternal and paternal groups. There are recent preprints which do this binning either using parents (Koren et al. 2018) or 10x+other datatypes (Chaisson et al. 2018). However, I didn't see anything in the methods which assigned reads to maternal or paternal haplotype. The Platanus graph is still built from both haplotypes and resolved afterwards, this should be clarified in the methods or the figure needs to be changed.
3. The method is introduced as "completely novel." Node linking and walking based on haplotype information is used by both supernova and FALCON-unzip so this statement needs more clarification. Which part of the algorithm is novel besides the integration of multiple data types.
4. The authors confound heterozygosity with repetitiveness. For example, when Supernova or FALCON-Unzip do not produce mega base scaffolds/contigs, this is ascribed to heterozygosity. The same statement is made about MP data, that it may be more useful in higher heterozygosity genomes. However, assemblies may be less continuous because of higher repeat content in the analyzed genomes or due to shorter molecule lengths. For example, the *A. thaliana* dataset used in the FALCON-Unzip paper is relatively heterozygous but still has multi-megabase contigs. The authors should use a simulated diploid with varying heterozygosity to control for genome complexity if they want to demonstrate this. There are simulators available for both PacBio and 10x datatypes.
5. There are gold-standard SNP sets (Eberle et al. 2017) as well as MHC typing results (Dilthey et al. 2016) for NA12878 that should be used for validation.
6. FALCON-Unzip should be included in the human validation results. The cited paper for its inferiority is the Supernova publication and so is not an independent evaluation. I expect FALCON-Unzip may capture some of the larger variation better than Supernova.

7. The inclusion of alternates in FALCON-Unzip is relatively well known. There are tools designed specifically to deal with this, for example `purge_haplotigs` (Roach et al. 2018). The authors should run `purge_haplotigs` on the FALCON-Unzip results and evaluate the post-purge primary assembly.

8. The FALCON-Unzip parameters have a very high length cutoff (15kb), how much coverage is over 15kb in the authors' dataset? The HPCdaligner options also seem out of date and don't match any parameters used in the FALCON-Unzip publication. How were these chosen? I'd suggest using parameters from the FALCON-Unzip publication instead. As presented now, I'm not sure the FALCON-Unzip assemblies are optimal.

9. Can the authors include the *A. thaliana* synthetic diploid genome from the FALCON-Unzip publication for validation (in a supplement or main text)? It has both PacBio and Illumina data and the assemblies are provided ensuring that FALCON-Unzip is run optimally.

10. The authors mention DRB5-like allele for NA12878, as far as I know one haplotype has DRB3 and the other does not, neither haplotype has DRB5. Could this text be clarified? The suggested MHC haplotype typing should confirm if the MHC is properly captured.

11. Is the source code for `platanus` available?

12. In terms of usability, I would suggest adding support for compressed files as well as `fasta/fastq` (that is `fastq.gz/fastq.bz2/etc`) so users don't have to extract all their data before running.

Responses to the reviewers

Article Title

Platanus-allee: a *de novo* haplotype assembler enabling a comprehensive access to divergent heterozygous regions

Authors

Rei Kajitani¹, Dai Yoshimura¹, Miki Okuno¹, Yohei Minakuchi², Hiroshi Kagoshima³, Asao Fujiyama³, Kaoru Kubokawa⁴, Yuji Kohara³, Atsushi Toyoda^{2,3}, Takehiko Itoh¹

¹ School of Life Science and Technology, Tokyo Institute of Technology, Meguro-ku, Tokyo, 152-8550, Japan

² Comparative Genomics Laboratory, National Institute of Genetics, Mishima, Shizuoka, 411-8540, Japan

³ Advanced Genomics Center, National Institute of Genetics, Mishima, Shizuoka, 411-8540, Japan

⁴ Research Center for Marine Education, Ocean Alliance, The University of Tokyo, Bunkyo-ku, Tokyo 113-0033, Japan

Correspondence should be addressed to T.I. (takehiko@bio.titech.ac.jp)

Note for the editor and all reviewers

We appreciate the valuable comments from the reviewers. During revision the manuscript has been extensively updated based on their suggestions. These changes have been described below along with point-by-point responses to the reviewers' comments. In the revised manuscript, changes are highlighted in gray.

Update of Platanus-allee

Following the first submission of the manuscript, two bugs in Platanus-allee (version 2.0.0) were discovered and fixed, and the software was updated as version 2.0.2. We have also updated the results in the manuscript, and the performance of Platanus-allee in several cases were improved. Note that the algorithm design was not modified, and the procedures described in the reviewed manuscript are truly implemented in the new version of Platanus-allee. The fixed bugs are as follows:

(1) Crash in the gap-closing module.

The "platanus_allee gap_close" command possibly crashes due to the wrong access to unallocated memory space. This bug was associated with handling of ambiguous bases ('N's) in input reads. The fixed version was named as 2.0.1.

(2) A bug in the untangling function.

This bug can cause untangling of cross structures in de Bruijn or scaffold graphs even if a condition to limit mis-untanglings is not satisfied. The corresponding one is "the links of the solution $\geq 4 \times n_{\text{alt-link}}$ " (see Methods; *Untangling cross structures in the (gapped) de Bruijn graph*; *Untangling cross structures in the scaffold graph*). The fixed version was named as 2.0.2.

Update of FALCON-Unzip

In the revised manuscript, FALCON-Unzip was updated to the newer version, binary from 11/02/2017, which was tested in another project about haplotype phasing^{R1}. In addition, multiple parameters were tested to optimize this tool and the result for human data was included.

Use of the additional post-assembly tools for FALCON-Unzip

As some reviewers have suggested, Illumina-reads-based polishing tool, Pilon^{R2}, and redundancy-reduction tools, Purge-Haplotigs (PH)^{R3}, were both applied for FALCON-Unzip and the results were added in the revised manuscript as "FALCON-Unzip, Pilon, PH".

Update of human PacBio data

To test the newer PacBio library type and execute FALCON-Unzip for the human genome (NA12878), we used P6-C4 reads deposited under BioProject PRJNA432857 instead of the previous P5-C3 reads.

Reviewer #1

Comment 1.1

On page 4 the authors state “The FALCON-Unzip15, the phasing tool for the single-molecule DNA sequencer, was used to construct long phased blocks for heterozygous plants and fungi, but the heterozygosities were not high.”- Could the authors give numbers to “not high” and state what they mean by “high”?

Response 1.1

We wanted to explain that the F1-hybrid *Arabidopsis thaliana* in the FALCON-Unzip paper^{R4} (estimated heterozygosity, ~1%) was not highly heterozygous compared to the organisms referred in the previous paragraphs. As the reviewer has pointed out, the sentence was not clear about what was compared and we instead described the detailed circumstance in the revised manuscript. (**page 4, line 52–61**).

In addition, we estimated heterozygosities of samples in this study based on *k*-mer frequency-information (**Supplementary Table 2**) and it is shown that the F1-hybrid *A. thaliana* is not "highly" heterozygous among these. Here, the heterozygosities of *P. polytes* (butterfly), *B. japonicum* (amphioxus) and *P. yedonensis* (cherry blossom, added in the revision) exceeded that of *A. thaliana*.

Comment 1.2

The authors state “Consequently, the development of a phasing tool able to handle these HDRs is one of the most urgent requirements for the comprehensive survey of diploid genome diversity.” - This may be overly superlative. I recommend “an important requirement” instead of “one of the most urgent requirements”

Response 1.2

We agree with the suggestion and have paraphrased the sentence in the manuscript (**page 5, line 77–78**).

Comment 1.3

Do I understand correctly from the supplementary tables 1-2 that the authors used only 20x PacBio coverage for Falcon-unzip? If so, this may be partly why it does not perform as well since I recall higher coverages were used in their paper.

Response 1.3

PacBio reads were downsampled to 20×-coverage only for Platanus-allee, while all reads were input into FALCON-Unzip. To emphasize this, we wrote "Platanus-allee input" for 20×-coverage PacBio reads in **Supplementary Table 1**, and expected coverage depths of PacBio reads were also written in many tables in the manuscript (e.g., "PacBio(x156)").

Comment 1.4

Did the authors use the recommended coverage for Supernova (I think older versions wanted ~50-60x? Which version of Supernova was used?

Response 1.4

The version of Supernova benchmarked was 2.0.0. As input read sets for Supernova, we tested all reads as well as reads downsampled to reach 56 of expected coverage depth, which is the optimum value in the manual, and selected a result that indicated larger scaffold-NG50 of phased-blocks (megabubbles). Although these were described in Supplementary Notes, Execution of Supernova, we added descriptions of these execution procedures before the benchmark results (**page 11, 172–173**).

Comment 1.5

I could not find where the authors made available the new data generated for this study. Could the authors clearly state this?

Response 1.5

We stated the SRA BioProject ID (PRJDB7193) of the data generated in this study in the "Data Availability" section. This entry will be open with the acceptance of this study.

To clarify whether a library was newly generated or not, we added "data generation" information in **Supplementary Table 1** (Statistics of sequenced reads).

Comment 1.6

I could not find the code for Platanus-allee. For this to have the highest impact, the code should be made publicly available with documentation prior to publication.

Response 1.6

We have made the source code freely available (<http://platanus.bio.titech.ac.jp/platanus2>) before submission of the manuscript for revision.

Reviewer #2

Comment 2.1

Your comparison is not totally fair. Falcon Unzip generates Contigs with Pacbio only and Supernova results in Scaffolds using 10x genomics reads only. Whereas Platanus-allele uses multiple technologies and even different libraries of illumina sequencing. For example, when you compare Platanus-allele using PE+ pacbio vs. Falcon Unzip you see that they both perform very similarly and in multiple cases Falcon Unzip outperforming Platanus (Table 1 and 2). Furthermore, you are not discussing the % of gabs or the total sum of gab bp in the assembly.

Response 2.1

We thank the reviewer for the comments. Performances of Platanus-allee such as NG50 are generally inferior to those of the others or it cannot be executed when only a 10X or PacBio library is used as input. In addition, combination of PE and PacBio without MP does not lead to a better result of Platanus-allee compared to that of FALCON-Unzip in many cases. However, Platanus-allee recorded the higher performances for several metrics using only Illumina libraries (PE + MPs) (**Table 1, 2; Fig. 2, 3, 4;** and so on) compared to the other tools including ones with high-coverage (>100×) PacBio libraries. In summary, it is difficult to run all tools for the same input and determine a ranking.

Basically, Platanus-allee is a hybrid assembler that primarily uses the accurate short-reads and can combine other types of libraries to enhance performances. We note that the design concept is to obtain the best quality of haplotype sequences utilizing all available libraries rather than reducing required inputs. In the revised manuscript, the purpose above was stated (**page 7–8, line 113–117**), and the paragraph discussing compositions of input library types including MPs was added to the Discussion section (**page 31–32, line 486–499**).

For the rate of gaps (Ns), we added descriptions for the points below: (1) Platanus-allee generated many gaps compared to the other tools, (2) especially FALCON-Unzip constructs completely gap-free sequences, (3) but Platanus-allee indicated better performances for recalls, and BUSCO-completeness for most cases. In addition, contig-NG50 values were added to the new tables and the differences between the tools are contrasted. Importantly, the precisions of Platanus-allee outperformed those of FALCON-

Unzip even when Pilon was used. We believe that the results of Platanus-allee have a certain value for practical uses. The discussion above was added in the manuscript (**page 32, line 502–510**).

Comment 2.2

I am also concerned that the 1kmer match strategy for the molecular reads for the evaluation is biased against the pacbio data set. Everyone uses illumina reads to improve the sequence consensus quality of the contigs and I don't think you used that before requiring a 100% match.

Response 2.2

As mentioned above, the Illumina-reads-based polishing tool, Pilon^{R2}, was applied and the results were included in the revised manuscript. Although the performances of recalls and precisions were improved after its application, the F-measures (harmonic mean of recall and precision) of Platanus-allee were never inferior to those on "FALCON-Unzip, Pilon, PH".

Comment 2.3

I am also a bit surprised about the low performance of Supernova even after scaffolding. Or did you used the contigs only?

Response 2.3

The N50 values of Supernova in the previous manuscript were measured for its scaffolds, not contigs. In the revision, contig-NG50 values were added and distinguished from scaffold-NG50 values. The reason of low performances of Supernova performances was inferred to be high heterozygosity from the simulation test of heterozygosity (**page 29, line 452–454; Supplementary Table 19, 20**).

Comment 2.4

I am also missing a discussion or table about the runtimes of each program.

Response 2.4

The run times of Platanus-allee under the condition of 20 threads were added as **Supplementary Table 6**. The corresponding paragraph was also added (**page 31, line 480–485**).

Comment 2.5

I would encourage you to discuss the cost vs. benefits for the different technology combinations Platanus-allele uses.

Response 2.5

About costs and features of sequencing technologies, we added the discussion of library configuration (**page 31–32, line 486–499**) and the paragraph about the costs (**page 33, 509–517**).

Comment 2.6

Have you tried to combine all technologies for Platanus ? So 10x + Pacbio + all illumina libraries?

Response 2.6

We tested inputting all library types (PE, MPs, 10X and PacBio) into Platanus-allee, and added these results to the revised manuscript. Although significant improvements were not observed in the highly heterozygous organisms (*P. polytes* and *B. japonicum*), these were effective for the human data, doubling scaffold-NG50 from that of PE+MPs+PacBio-input and archiving the top F-measure in Moleculo-1k-mer evaluation (**Fig. 4a, Supplementary Table 11**).

Comment 2.7

What confuses me a bit is the question if Platanus is optimized only for these highly

heterozygous crosses/ animals and thus its utility to more general genomes. Maybe you can comment on this a bit more in the discussion since the results are quite different between human and the eg. Butterfly. What would you expect in terms of plants, or other mammals?

Response 2.7

Platanus-allee is designed to exactly distinguish haplotypes utilizing accurate short-reads (Illumina) in the phasing step, and we expect that this function is effective to a wide range of diploid organisms, not only to "exceptional" ones. With Illumina MP libraries, Platanus-allee will have more advantage against the other tools for the highly heterozygous samples. To quantify heterozygosity, we applied the *k*-mer analysis tool, GenomeScope^{R5}, which can estimate that value, to all species tested in this study (**Supplementary Table 2**). Also, we added three samples: *A. thaliana* F1-hybrid^{R4}, a cherry blossom^{R6} and a bat^{R7}, which have public sequencing data (**page 26–28, line 401–438; Supplementary Table 16**). If MP libraries were available, Platanus-allee exhibited large scaffold-NG50 values (≥ 290 kbp) using PE + MPs for all samples except for human (**Supplementary Table 17**). In this study, Platanus-allee indicated effectiveness for samples with heterozygosities ranging 0.38–3.48% compared with the other tools, inferring its versatility.

Comment 2.8

I would also encourage you to include the comparison to Platanus the previous version of the de novo assembly software to highlight the improvements over this implementation.

Response 2.8

We have now included the results of Platanus (version 1) in the tables for consensus sequences (*e.g.*, **Table 2**). Platanus cannot construct phased blocks (*i.e.*, each haplotype sequences) and is not included in many tables and figures for phased blocks. Before showing the results, we have added a sentence to explain the difference of the functions between these tools (**page 11–12, line 175–177**).

Reviewer #3

Comment 3.1

The tables present N50 stats but it is not clear if these are contig or scaffold statistics. Based on the stats, these appear to be scaffolds. However, FALCON-Unzip only produces contigs so this comparison is not fair. It is also not clear if the genome size is kept constant between different assemblies. The statistics using a consistent genome size and contigs should be presented for all assemblers.

Response 3.1

In the revised manuscript, we have used scaffold-NG50 and contig-NG50 instead of "N50". We defined "scaffold-NG50" equals contig-NG50 if an assembled sequence has no gaps, such as the FALCON-Unzip's result. The description of the indicators above was added before the benchmark results (**page 10–11, line 156–161**).

Comment 3.2

Figure 1 panel a is not clear to me. The figure shows the reads being binned by maternal and paternal groups. There are recent preprints which do this binning either using parents (Koren et al. 2018) or 10x+other datatypes (Chaisson et al. 2018). However, I didn't see anything in the methods which assigned reads to maternal or paternal haplotype. The Platanus graph is still built from both haplotypes and resolved afterwards, this should be clarified in the methods or the figure needs to be changed.

Response 3.2

We thank the reviewer for this observation and have modified **Fig. 1** (schematic model of phasing and haplotype synteny-based assembly) so that pre-assembly binning of reads is not implied. In the new version, the actual functions of Platanus-allee are depicted more clearly, focusing on the de Bruijn graph.

Comment 3.3

The method is introduced as "completely novel." Node linking and walking based on

haplotype information is used by both supernova and FALCON-unzip so this statement needs more clarification. Which part of the algorithm is novel besides the integration of multiple data types.

Response 3.3

To elucidate the features of the Platanus-allee algorithm, we have added more detailed explanations in the Results section, under Development of Platanus-allee (**page 6, line 90–107**). In contrast to the existing tools, the feature of Platanus-allee is that it never constructs consensus sequences by multiple alignments or removal of one side of bubble structures in graphs before finishing phasing. We have explained the difference between Platanus-allee and the other tools and the relations to the input reads in the revised text.

Comment 3.4

The authors confound heterozygosity with repetitiveness. For example, when Supernova or FALCON-Unzip do not produce mega base scaffolds/contigs, this is ascribed to heterozygosity. The same statement is made about MP data, that it may be more useful in higher heterozygosity genomes. However, assemblies may be less continuous because of higher repeat content in the analyzed genomes or due to shorter molecule lengths. For example, the *A. thaliana* dataset used in the FALCON-Unzip paper is relatively heterozygous but still has multi-megabase contigs. The authors should use a simulated diploid with varying heterozygosity to control for genome complexity if they want to demonstrate this. There are simulators available for both PacBio and 10x datatypes.

Response 3.4

As the reviewer suggested, we performed the simulation benchmark and add the section of Results, Benchmarking for fully simulated heterozygous data (**page 28–29, line 440–457**). The results are shown in **Supplementary Table 19, 20**. Although most NG50 values outperform those of the actual data possibly unknown natures of actual ones, there are two important tendencies:

(1) The performance of Supernova extremely became worse for 2%-heterozygous data. This is consistent with the bad performances of the tool for *P. polytes* and *B. japonicum*,

implying that it cannot handle high heterozygosity (>1%).

(2) Platanus-allee exceeded FALCON-Unzip even with Pilon and PH when heterozygosity $\geq 1\%$ for most indicators. This supports the advantage of Platanus-allee for HDRs.

Comment 3.5

There are gold-standard SNP sets (Eberle et al. 2017) as well as MHC typing results (Dilthey et al. 2016) for NA12878 that should be used for validation.

Response 3.5

We performed benchmarks using the two suggested data sets^{R8,R9} and added the results to the section of the benchmark using human NA12878 data (**page 22–23, line 348–361**). The detailed values were added as **Supplementary Table 12, 13 and 14**, and the summary is given below.

For Recalls and precisions measured based on fixed-length fragments from the haplotypes of the Platinum SNP set^{R8} (**Supplementary Table 12**), the trend was consistent with those for the Moleculo contigs.

Using the previous typing results of six loci in the MHC region^{R9}, we counted the loci correctly represented in the phased blocks. As a result (**Supplementary Table 14**), Platanus-allee with MP and/or PacBio and FALCON-Unzip correctly typed all loci, and the number of exact-matches to database-sequences was the most for Platanus-allee, supporting that this tool has an advantage to phase the MHC region.

Comment 3.6

FALCON-Unzip should be included in the human validation results. The cited paper for its inferiority is the Supernova publication and so is not an independent evaluation. I expect FALCON-Unzip may capture some of the larger variation better than Supernova.

Response 3.6

We have benchmarked FALCON-Unzip for the human NA12878 data and its results were included in the revised manuscript. As the reviewer predicted, it exhibited high

performances for some divergent regions such as MHC region.

Comment 3.7

The inclusion of alternates in FALCON-Unzip is relatively well known. There are tools designed specifically to deal with this, for example `purge_haplotigs` (Roach et al. 2018). The authors should run `purge_haplotigs` on the FALCON-Unzip results and evaluate the post-purge primary assembly.

Response 3.7

Following the reviewer's suggestion, we have applied the post-assembly tools, `Purge-Haplotigs` (PH) and `Pilon` (related with **Comment 2.2**), to the results of FALCON-Unzip, and added the final results to the revised manuscript.

Comment 3.8

The FALCON-Unzip parameters have a very high length cutoff (15kb), how much coverage is over 15kb in the authors' dataset? The `HPCdaligner` options also seem out of date and don't match any parameters used in the FALCON-Unzip publication. How were these chosen? I'd suggest using parameters from the FALCON-Unzip publication instead. As presented now, I'm not sure the FALCON-Unzip assemblies are optimal.

Response 3.8

In this revision, to optimize FALCON-Unzip, we have increased the number of trials for the parameter sets and have elected the result indicating the maximum primary-contig-NG50 for each sample (**Supplementary Table 3**). The parameter sets are as follows:

(1) FALCON-integrate-based

This had been used in the previous manuscript and based on the example config file included in the FALCON-integrate package (<https://github.com/PacificBiosciences/FALCON-integrate>). Although the read-length cutoff is 15 kbp, which was high compared to others as the reviewer had pointed out, the expected coverage depth ranged from 43× to 107× for reads whose length ≥ 15 kbp and

led to the best NG50s for the two species (*B. japonicum* and *C. elegans*). The parameters modified from the base file except for parallelization (the number of threads and so on) were as follows:

(i) -s of "pa_HPCdaligner_option" and "ovlp_HPCdaligner_option" (1000 → 100)

This was modified to avoid failures of processes, based on the suggestion from the developer. in the issue-page

(<https://github.com/PacificBiosciences/FALCON/issues/444>).

(ii) -t of "ovlp_HPCdaligner_option" (32 → 16)

This was modified to reduce memory-usage and avoid failures. The value was determined based on the instruction in the README of DALIGNER (<https://github.com/thegenemyers/DALIGNER>) and other example config files in the web page of FALCON

(<https://pb-falcon.readthedocs.io/en/latest/parameters.html#parameters>).

(iii) skip_checks = True

This was added to avoid failures related to the file-system based on the comment from the developer in the issue-page

(<https://github.com/PacificBiosciences/FALCON/issues/451>).

(2) Chin et al. 2016-based

This is the parameter set used in the original paper of FALCON-Unzip^{R4}, that the reviewer had suggested. The read-length cutoff is 4 kbp.

(3) Koren et al. 2018-based

This was used in the previous study^{R1} to assemble the human NA12878 data. In addition to the original value of read-length cutoff (5 kbp), 4 kbp, which was suggested by the reviewer, was also tested.

The detailed config files were described in the section of Supplementary Note Execution of FALCON-Unzip.

Comment 3.9

Can the authors include the *A. thaliana* synthetic diploid genome from the FALCON-Unzip publication for validation (in a supplement or main text)? It has both PacBio and Illumina data and the assemblies are provided ensuring that FALCON-Unzip is run optimally.

Response 3.9

We have performed the benchmark for *A. thaliana* F1-hybrid^{R4} (**page 26–27, line 405–414; Supplementary Table 17–19**), utilizing the assemblies of the parental strains as reference genomes. The heterozygosity of this sample was estimated as 1.05% from *k*-mer frequency information (tool, GenomeScope^{R5}) (**Supplementary Table 2**), which fell between the values of highly heterozygous samples (*P. polytes* and *B. japonicum*) and those of lowly heterozygous ones (*C. elegans* and *H. sapiens*), in this study. Probably because MP libraries were not available, the performances of Platanus-allee were generally poor, which is consistent with the benchmarks of the other samples. FALCON-Unzip outperformed Platanus-allee for all indicators of phased blocks except the precision of 1k-mers (**Supplementary Table 17, 19**).

We also performed the benchmark using the plant sample with MP libraries added in the revision, *P. yedonensis* (cherry blossom), and further discussed the versatility of the tool (**page 27, line 416–421**). We suspect that the high performance of FALCON-Unzip for the *A. thaliana* F1-hybrid is a rare case. In summary, the added benchmarks confirm the tendency of Platanus-allee about input libraries and illustrates the fluctuation of the FALCON-Unzip performances.

Comment 3.10

The authors mention DRB5-like allele for NA12878, as far as I know one haplotype has DBR3 and the other does not, neither haplotype has DRB5. Could this text be clarified? The suggested MHC haplotype typing should confirm if the MHC is properly captured.

Response 3.10

We thank the reviewer for the comment. The word "*HLA-DRB5*" in the previous manuscript indicated the locus name on the primary reference genome (GRCh38), and

we have confirmed that the *HLA-DRB5* gene is not found (identity $\geq 95\%$ and alignment-coverage $\geq 50\%$) in any of the assemblies. In addition, *HLA-DRB3* gene was found in one of the haplotype Platanus-allee constructed, which was concordant with the report based on the trio (father, mother and offspring) data and PacBio reads^{R1}. We have added the above results to the section of the benchmarks using human data (**page 24, line 377–380**) and **Fig. 4d**.

Comment 3.11

Is the source code for platanus available?

Response 3.11

We have made the source code freely available (<http://platanus.bio.titech.ac.jp/platanus2>) at the same time of the submission of the revision.

Comment 3.12

In terms of usability, I would suggest adding support for compressed files as well as fasta/fastq (that is fastq.gz/fastq.bz2/etc) so users don't have to extract all their data before running.

Response 3.12

We have added the function to accept gzip or bzip2-compressed files to Platanus-allee v2.0.2, which is tested in the revision. File formats have been automatically detected.

References

- R1. Koren, S. *et al.* *De novo* assembly of haplotype-resolved genomes with trio binning. **36**, 1174–1182 (2018).
- R2. Walker, B. J. *et al.* Pilon: An Integrated Tool for Comprehensive Microbial Variant Detection and Genome Assembly Improvement. *PLoS One* **9**, e112963 (2014).
- R3. Roach, M. J., Schmidt, S. A. & Borneman, A. R. Purge Haplotigs: allelic contig reassignment for third-gen diploid genome assemblies. *BMC Bioinformatics* **19**, 460 (2018).
- R4. Chin, C.-S. *et al.* Phased diploid genome assembly with single-molecule real-time sequencing. *Nat. Methods* **13**, 1050–1054 (2016).
- R5. Vurture, G. W. *et al.* GenomeScope: fast reference-free genome profiling from short reads. *Bioinformatics* **33**, 2202–2204 (2017).
- R6. Baek, S. *et al.* Draft genome sequence of wild *Prunus yedoensis* reveals massive inter-specific hybridization between sympatric flowering cherries. *Genome Biol.* **19**, 1–17 (2018).
- R7. Zhang, G. *et al.* Comparative analysis of bat genomes provides insight into the evolution of flight and immunity. *Science* **339**, 456–460 (2013).
- R8. Eberle, MA. *et al.* A reference dataset of 5.4 million phased human variants validated by genetic inheritance from sequencing a three-generation 17-member pedigree. *Genome Res.* **27**, 157–164 (2017).
- R9. Dilthey, A. T. *et al.* High-Accuracy HLA Type Inference from Whole-Genome Sequencing Data Using Population Reference Graphs. *PLoS Comput. Biol.* **12**, e1005151 (2016).

Reviewer #1 (Remarks to the Author):

The authors have done a good job responding to the reviewers' comments, and I have no further recommendations.

Reviewer #2 (Remarks to the Author):

The authors of the manuscript addressed all my concerns and questions and updated the manuscript accordingly.

Reviewer #3 (Remarks to the Author):

The authors have addressed the majority of my concerns. I had a few remaining comments:

1. I am still not clear on the difference with prior phasing algorithms. It is true that FALCON-Unzip first makes a collapsed assembly and then relies on sequence mapping to phase. However, as far as I understand, Supernova's line subgraphs are not linear collapses but preserve the bubbles from variants within the graph. It seems this strategy is quite similar to that used by Platanus, albeit with Platanus supporting more datatypes.

2. Figures 2b, 2e, 3b, 4c, I presume the plots are showing scaffolds for Supernova and Platanus-allee but contigs for Falcon-unzip? I would suggest indicating gap locations in the scaffolds (as is done in 4c for one large gap in the supernova scaffolds but I presume this is not the only gap). This would better indicate which bases are unknown in the assemblies.

3. The authors state that the FALCON-unzip recipe and TrioBinning are expensive. Both should be dominated by the PacBio cost since the PE libraries required for trio binning are negligible in cost by the authors discussion. I expect the cost of MP libraries is dominated by library prep not sequencing whereas generating PacBio data is dominated by sequencing costs. This would mean the tradeoff in cost likely depends on the genome size, with larger genomes becoming more expensive with FALCON-Unzip due to more required PacBio sequencing. On small genomes, generating 4+ MP libraries may actually cost more than moderate PacBio sequencing. Can the authors comment on this?

Minor:

1. Table 1, some numbers are not visible due to column width (e.g. falcon unzip result for B. japonicum replaced with #####).

2. line 223 scaffold misspelled

Responses to the reviewers (the second round)

Article Title

Platanus-allee: a *de novo* haplotype assembler enabling a comprehensive access to divergent heterozygous regions

Authors

Rei Kajitani¹, Dai Yoshimura¹, Miki Okuno¹, Yohei Minakuchi², Hiroshi Kagoshima³, Asao Fujiyama³, Kaoru Kubokawa⁴, Yuji Kohara³, Atsushi Toyoda^{2,3}, Takehiko Itoh¹

¹ School of Life Science and Technology, Tokyo Institute of Technology, Meguro-ku, Tokyo, 152-8550, Japan

² Comparative Genomics Laboratory, National Institute of Genetics, Mishima, Shizuoka, 411-8540, Japan

³ Advanced Genomics Center, National Institute of Genetics, Mishima, Shizuoka, 411-8540, Japan

⁴ Research Center for Marine Education, Ocean Alliance, The University of Tokyo, Bunkyo-ku, Tokyo 113-0033, Japan

Correspondence should be addressed to T.I. (takehiko@bio.titech.ac.jp)

Note for the editor and all reviewers

We thank the editor and the reviewers for the second round of review. In the revised manuscript, changes are highlighted in gray.

Reviewer #3

Comment 3.1

I am still not clear on the difference with prior phasing algorithms. It is true that FALCON-Unzip first makes a collapsed assembly and then relies on sequence mapping to phase. However, as far as I understand, Supernova's line subgraphs are not linear collapses but preserve the bubbles from variants within the graph. It seems this strategy is quite similar to that used by Platanus, albeit with Platanus supporting more datatypes.

Response 3.1

We have added a more detailed explanation (**page 6–7, line 94–102**). As the reviewer pointed out, Supernova retains bubbles in a de Bruijn graph and detects the linkages between them in the final phasing step. However, in the Supernova workflow, the assembly functions such as scaffolding and gap-closing are not used to extend each haplotype independently, in contrast to Platanus-allee.

In fact, some procedures of Supernova are unclear especially for the version 2. Therefore the phrase "To our knowledge from the original description of Supernova" have been added (**page 6, line 94–95**). Nonetheless, the differences between Supernova and Platanus-allee were obvious for the performances for HDRs and we emphasize this in the discussion section (**page 31, line 487–490**).

Comment 3.2

Figures 2b, 2e, 3b, 4c, I presume the plots are showing scaffolds for Supernova and Platanus-allee but contigs for Falcon-unzip? I would suggest indicating gap locations in the scaffolds (as is done in 4c for one large gap in the supernova scaffolds but I presume this is not the only gap). This would better indicate which bases are unknown in the

assemblies.

Response 3.2

As the reviewer suggested, the figures (2b, 2e, 3b, 4c, and 4e) have been revised and gap regions ('N's) are represented as gray lines. We assume that these figures have become more informative.

Comment 3.3

The authors state that the FALCON-unzip recipe and TrioBinning are expensive. Both should be dominated by the PacBio cost since the PE libraries required for trio binning are negligible in cost by the authors discussion. I expect the cost of MP libraries is dominated by library prep not sequencing whereas generating PacBio data is dominated by sequencing costs. This would mean the tradeoff in cost likely depends on the genome size, with larger genomes becoming more expensive with FALCON-Unzip due to more required PacBio sequencing. On small genomes, generating 4+ MP libraries may actually cost more than moderate PacBio sequencing. Can the authors comment on this?

Response 3.3

As the reviewer pointed out, the efficiency of costs may vary according to genome sizes, and we have added the phrase "although the relation is possibly reversed for the samples with smaller genome sizes" (**page 34, line 526–527**).

We found the new study^{R1} comparing the costs of the library-preparations and the sequencing technologies, which supports our estimation, and added this to the manuscript (**page 33, line 522**). Note that the basic structures of the discussion section have not been changed.

Comment 3.4

Minor:

1. Table 1, some numbers are not visible due to column width (e.g. falcon unzip result for *B. japonicum* replaced with #####).

2. line 223 scaffold misspelled

Response 3.4

We have fixed these points and thank the careful checking.

References

R1. Paajanen, P. *et al.* A critical comparison of technologies for a plant genome sequencing project. *Gigascience* doi:10.1093/gigascience/giy163 (2019).